# Promoter-proximal pausing mediated by the exon junction complex regulates splicing

Junaid Akhtar[1,5], Nastasja Kreim[2], Federico Marini [3], Giriram Mohana[1], Daniel Brüne[1], Harald Binder[4] & Jean-Yves Roignant[1]

Promoter-proximal pausing of RNA polymerase II (Pol II) is a widespread transcriptional regulatory step across metazoans. Here we find that the nuclear exon junction complex (pre-EJC) is a critical and conserved regulator of this process. Depletion of pre-EJC subunits leads to a global decrease in Pol II pausing and to premature entry into elongation. This effect occurs, at least in part, via non-canonical recruitment of pre-EJC components at promoters. Failure to recruit the pre-EJC at promoters results in increased binding of the positive transcription elongation complex (P-TEFb) and in enhanced Pol II release. Notably, restoring pausing is sufficient to rescue exon skipping and the photoreceptor differentiation defect associated with depletion of pre-EJC components in vivo. We propose that the pre-EJC serves as an early transcriptional checkpoint to prevent premature entry into elongation, ensuring proper recruitment of RNA processing components that are necessary for exon definition.

---

[1] Laboratory of RNA Epigenetics, Institute of Molecular Biology (IMB), 55128 Mainz, Germany. [2] Bioinformatics Core Facility, Institute of Molecular Biology (IMB), 55128 Mainz, Germany. [3] Institute of Medical Biostatistics, Epidemiology and Informatics (IMBEI), 55101 Mainz, Germany. [4] Institute of Medical Biometry and Statistics, Faculty of Medicine and Medical Center, University of Freiburg, 79104 Freiburg, Germany. [5] Present address: Institute of Neurobiology and Developmental Biology, JGU, 55128 Mainz, Germany. Correspondence and requests for materials should be addressed to J.-Y.R. (email: j.roignant@imb-mainz.de)

Transcripts produced by RNA polymerase II (Pol II) undergo several modifications before being translated, including 5′-end capping, intron removal, 3′-end cleavage and polyadenylation. These events usually initiate co-transcriptionally while the nascent transcript is still tethered to the DNA by Pol II[1–4]. This temporal overlap is important for the coupling between these processes[5–9]. Initially, Pol II is found in a hypophosphorylated form at promoters. At the onset of initiation, the CTD of Pol II becomes phosphorylated at the Ser5 position. Pol II subsequently elongates and often stalls 20–60 nucleotides downstream of transcription start sites (TSS), an event commonly referred to promoter proximal pausing[10,11]. Promoter proximal pausing of Pol II is widely seen at developmentally regulated genes, and is thought to play critical roles in facilitating rapid and synchronous transcriptional activity upon stimulation[12–17]. Pol II pausing is also suggested to act as a checkpoint influencing downstream RNA processing events such as capping and splicing, but evidence for this function is still limited. The transition from the paused state to elongation is promoted by the positive tran-scription elongation factor (P-TEFb) complex, which includes the cyclin-dependent kinase 9 (Cdk9) and cyclin T[18–21]. P-TEFb phosphorylates Ser2 of the CTD as well as the negative elongation factor (NELF) and DRB sensitivity-inducing factor (DSIF), leading to the release of Pol II from promoter[22–24]. Another related kinase, Cdk12, was also recently suggested to affect Pol II pausing after its recruitment through Pol II-associated factor 1 (PAF1)[25,26].

The exon junction complex (EJC) is a ribonucleoprotein complex, which assembles on RNA upstream of exon-exon boundaries as a consequence of pre-mRNA splicing[27,28]. The spliceosome-associated factor CWC22 is essential to initiate this recruitment[29–32]. The nuclear EJC core complex, also called pre-EJC, is composed of the DEAD box RNA helicase eIF4AIII[33], the heterodimer Mago nashi (Mago)[34] and Tsunagi (Tsu/Y14)[35,36]. The last core component, Barentsz (Btz), joins and stabilizes the complex during or after export of the RNA to the cytoplasm[37]. Non-canonical association of Y14 at promoters has also been previously reported, although the significance of this binding remains unknown[38]. The EJC has been shown to play crucial roles in post-transcriptional events such as RNA localization, translation and nonsense-mediated decay[39–41]. These functions are mediated by transient interactions of the core complex with effector proteins[42].

The pre-EJC, along with the accessory factors RnpS1 and Acinus, participate in intron definition[43,44]. In absence of the pre-EJC, many introns containing weak splice sites are retained. The pre-EJC facilitates removal of weak introns by a mechanism involving its prior deposition to adjacent exon junctions. In addition, the depletion of pre-EJC components results in frequent exon-skipping events, particularly at large intron-containing transcripts, although the mechanism is poorly understood[45–47]. In Drosophila, loss of Mago in the eye leads to several exon skipping in MAPK, resulting in photoreceptor differentiation defects. Other large transcripts, often expressed from hetero-chromatic regions, show the same Mago-splicing dependency. Similarly, in human, exons flanked by longer introns are more dependent on the EJC for their splicing[47].

Here, we investigated the mechanism underlying the role of the pre-EJC in exon definition in Drosophila. We observed that depletion of pre-EJC components, but not of the EJC splicing subunit RnpS1, lead to a global decrease in promoter proximal pausing, altered Pol II phosphorylation state and premature entry into elongation. These changes are concomitant with underlying changes in chromatin architecture and correlate strongly with exon skipping events. These effects are driven by non-canonical recruitment of pre-EJC components at promoters. Co-

immunoprecipitation experiments indicated that Mago associates with Pol II but this association is largely dependent on nascent RNA. Upon knockdown (KD) of pre-EJC components, Cdk9 binding to Pol II is increased, partly accounting for the premature Pol II release. Remarkably, genetically increasing Pol II pausing rescues exon skipping events and the eye phenotype associated with KD of pre-EJC components, indicating that restraining Pol II release into gene bodies is sufficient to com-plement the loss of pre-EJC components in exon definition. Altogether, our results demonstrate a direct role of the pre-EJC in exon definition via the control of promoter proximal pausing.

## Results

**The pre-EJC regulates expression of long genes**. To investigate the role of the EJC in exon definition, we performed RNAi in Drosophila S2R+ cells. As expected, Mago depletion triggered exon skipping in MAPK in Drosophila cells (Supplementary Figure 1a-c)[45,46]. Further, we found that depletion of other pre-EJC components (eIF4AIII and Y14), but not of the cytoplasmic EJC subunit Btz or the accessory factor RnpS1, strongly impaired MAPK splicing and expression of large-intron containing tran-scripts (Supplementary Figure 1a–c, f, g). In particular, depletion of pre-EJC components led to a higher number of exon skipping events than depletion of Btz or RnpS1 (Supplementary Figure 1h and data not shown). This effect requires pre-EJC assembly as a mutant version of Mago, which is unable to bind Y14, failed to rescue the MAPK splicing defect (Supplementary Figure 1d, e). Thus, the pre-EJC is required for proper expression and splicing of large intron-containing genes. In contrast to intron definition, this exon definition activity only slightly required the EJC splicing subunit RnpS1, suggesting a distinct mechanism.

**Lack of pre-EJC alters Pol II phosphorylation**. Introns are spliced while nascent RNA is still tethered to Pol II, allowing coupling between splicing and transcription machineries[6,7,9,48–50]. To address whether the pre-EJC regulates splicing via modulation of transcription, we performed chromatin immunoprecipitation (ChIP) experiments for the different forms of Pol II. We found that Mago KD results in decrease of total Pol II occupancy at the 5′ end of MAPK while the distribution in the rest of the gene body was comparable to the control (Fig. 1a). In addition, the elon-gating Ser2-phosphorylated (Ser2P) form of Pol II was mildly decreased at the TSS, but significantly enriched along the gene body (Fig. 1a). This was specific to Mago depletion and on pre-EJC assembly, as reintroducing WT Mago cDNA, unlike the mutant version, restores the wild type profiles (Supplementary Figure 2a, b). Examining Pol II and Ser2P profiles in a genome-wide manner show extensive changes with decrease at the TSS and increase towards transcription end sites (TES) (Fig. 1b–e and supplementary Figure 2c). Similar changes in Pol II occupancy were observed upon depletion of Y14 and eIF4AIII, especially at the TSS (Supplementary Figure 2d–f), but neither on depletion of RnPS1 nor of Btz (Supplementary Figure 2d–h). Thus, pre-EJC components regulate Pol II distribution genome-wide.

**The pre-EJC facilitates pol II pausing**. To further investigate transcriptional changes in pre-EJC-depleted cells, we analyzed the Pol II release ratio (PRR), which is the ratio of Pol II occupancy between gene bodies and promoter regions (Fig. 1f). Notably, depletion of pre-EJC components, but not of RnpS1, significantly increased the PRR (Fig. 1g and Supplementary Figure 2i,j). Together, these results indicate an unanticipated and specific role for pre-EJC components in promoting promoter-proximal pausing of RNA Pol II. We next divided the changes in PRR into four equal size quartiles, from low to high PRR derived from

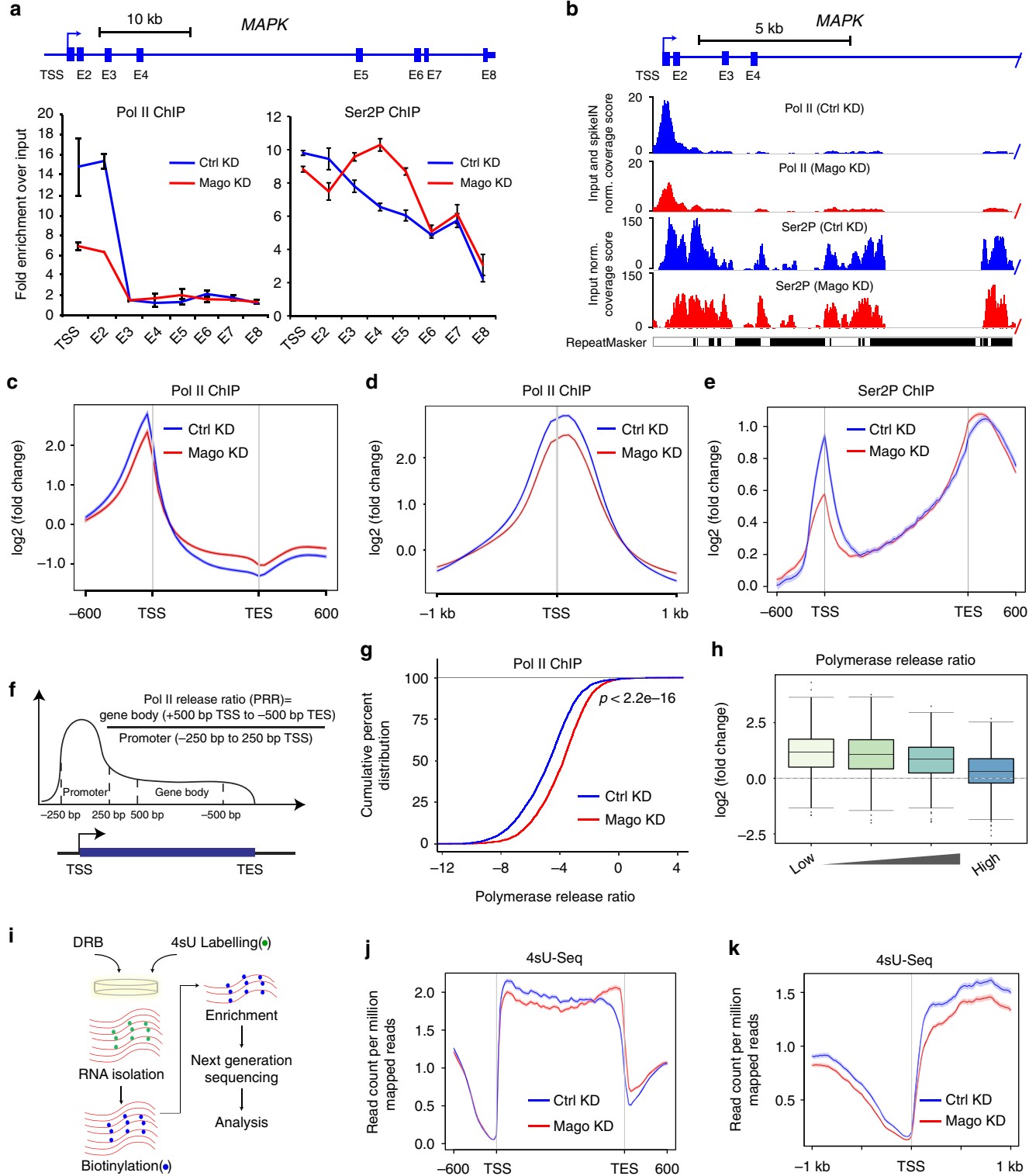

Pol II occupancy in WT condition. When classified accordingly, the quartile with the lowest PRRs showed largest increase in PRR upon Mago KD (Fig. 1h), suggesting that strongly paused genes are more affected upon the loss of the pre-EJC.

Next, we performed 4sU-seq to detect nascent transcripts (a modified approach of[51], see Material and methods), (Fig. 1i). The 4sU-Seq metagene profile of Mago KD cells revealed lower read counts at the TSS and an increase towards the 3′ end of transcripts, consistent with the Pol II ChIP-Seq and reduced pausing (Fig. 1j, k). To monitor nascent transcription overtime,

we coupled this approach to treatment with the pausing inhibitor 5,6- dichlorobenzimidazole 1-β-D-ribofuranoide (DRB), (4sU-DRB-seq)[52,53]. Our analysis revealed an average elongation rate of 1 kb per minute in *Drosophila* S2 cells, in agreement with previous reports but slower than in human cells (Supplementary Figure 3a–c)[54–57]. Importantly, in contrast to the widespread change in promoter-proximal pausing, the average elongation rate was unaffected in Mago-depleted cells (Supplementary Fig. 3c), and the moderate gene-to-gene variation in elongation rate did not correlate with changes in exon inclusion

**Fig. 1** Mago prevents premature release into transcription elongation. **a** ChIP-qPCR analysis of Pol II and Ser2P occupancies at *MAPK* locus. The tested regions for enrichment are shown in the scheme. Error bars indicate the standard deviation of three biological replicates. **b** Track examples of total Pol II and Ser2P ChIP-Seq from S2R+ cells extracts, after either control or Mago knockdown. The tracks are average of two independent biological replicates after input and "spike-in" normalization. Shown here are the profiles on *MAPK*, a well-described pre-EJC target gene. **c**, **d** Metagene profiles of averaged total Pol II occupancies from two independent biological replicates after "spike-in" normalization in control and Mago-depleted cells along with standard error of mean for all the expressed genes, −600 bp upstream of transcription start sites (TSS) and +600 bp downstream of transcription end sites (TES) (**c**); or centered at the TSS in a ±1 Kb window (**d**). Log2 fold changes against input control are shown on *Y*-axis, while *X*-axis depicts genomic coordinates. **e** Metagene profiles of averaged Ser2P occupancies in control and Mago-depleted cells of two independent biological replicates also displaying standard error of mean for all expressed genes. Log2 fold changes against input control are shown on *Y*-axis, while *X*-axis depicts scaled genomic coordinates. **f** Schematic representation of the calculation of the Pol II release ratio (PRR). The promoter is defined as 250 bp upstream and downstream of TSS, while the gene body is 500 bp downstream of TSS to 500 bp upstream of transcription end site (TES). **g** The empirical cumulative distribution function (ECDF) plot of computed PRR in control and Mago knockdown conditions, after "spike-in" normalization. *p*-value is derived from two-sample Kolmogorov-Smirnov test. **h** Box plots showing changes in PRRs upon Mago depletion when compared to control, separated into different PRR quartiles. The quartiles were generated for genes that are Pol II bound in both control and Mago knockdown conditions. **i** Schematic depiction of the DRB-4sU-Seq approach. **j** Metagene profile of nascent RNA from non-DRB treated 4sU-Seq data in control and Mago-depleted cells, with standard error of mean for all the expressed genes. Averaged read counts per million of mapped reads of two independent biological replicates from 4sU-Seq are shown on *Y*-axis while *X*-axis depicts scaled genomic coordinates. **k** Metagene profile of nascent RNA from non-DRB treated 4sU-Seq data in control and Mago-depleted cells with standard error of mean for all the expressed genes based on the average of two independent biological replicates, centered at the TSS in a ±1 Kb window. Nascent RNA was fragmented to ≤100 bp during enrichment. Averaged read counts per million of mapped reads of two independent biological replicates from 4sU-Seq are shown on *Y*-axis while *X*-axis depicts genomic coordinates

(Supplementary Figure 3d). Altogether, our data suggest that the pre-EJC controls Pol II pausing but does not significantly affect the elongation rate.

To better dissect the role of pre-EJC in Pol II pausing we examined *heat shock* (*hsp*) genes, which possess a promoter-proximal Pol II that has been extensively characterized[58]. We performed Pol II ChIP-qPCR to monitor Pol II occupancy before, during and after HS on the *Hsp70Aa* gene. We found that Pol II occupancy at the 5′ end of the gene was higher in control cells compared to Mago-depleted cells before HS (Supplementary Figure 4). During HS, Pol II occupancy rose dramatically and the extent of this increase was similar in control versus Mago KD, suggesting that Mago has no impact on transcription initiation. However, during recovery after HS, Pol II occupancy remained high at the 5′ end of the gene in control condition but was significantly lower in the Mago KD. These results thus further suggest that the pre-EJC is specifically involved in the control of Pol II pausing rather than in transcription initiation.

**Pre-EJC components associate at promoter regions**. To investigate how the pre-EJC controls Pol II pausing, first we evaluated the expression of known pausing factors. Depletion of pre-EJC components did not affect the expression of Cdk9, Spt5, subunits of the NELF complex, GAGA, Med26, TFIID (Supplementary Figure 5a–d). To test whether the pre-EJC might itself associate with chromatin, as suggested by previous immunostaining of pre-EJC components on polytene chromosomes of *Drosophila* salivary glands[38], we performed ChIP-Seq experiments. We observed genome-wide enrichment of HA-tagged pre-EJC components, but not of RnpS1-HA, primarily at promoters of expressed genes (Fig. 2a–c and Supplementary Figure 6a, c). Mago depletion reduced the Mago-HA enrichment, demonstrating the specificity of the signal (Supplementary Figure 6c). The degree of overlap between the bound targets of pre-EJC components was 34%, corresponding to 816 genes (Supplementary Figure 6d).

Next, we tested whether Mago might associate with promoters via an interaction with RNA Pol II. We found that Flag-tagged Mago bound to Pol II by co-IP (Fig. 2d). Importantly, FLAG-Mago interacted with Pol II Ser5P but not with elongating Ser2P, potentially explaining the enrichment of pre-EJC binding at promoter regions. In addition, interaction with Pol II was reduced after treatment with RNase T1, indicating that a RNA intermediate facilitates this association (Fig. 2d). Accordingly,

most of Mago binding to promoters was lost when the chromatin was treated with RNAse T1 prior to immunoprecipitation (Fig. 2e and Supplementary Figure 6a, 6c). Furthermore, only mRNAs whose corresponding genes were bound by pre-EJC co-immunoprecipitated with Mago-HA, including intronless transcripts, indicating that in contrast to canonical EJC deposition, association of Mago at promoters can occur independently of pre-mRNA splicing (Supplementary Figure 6e). Further, the association of Mago-HA with the TSS was not substantially affected by the depletion of the spliceosome-associated factor CWC22 (Supplementary Figure 6f, g) or by treatment with a splicing inhibitor on intronless genes. Nevertheless, applying the same condition on intron-containing genes significantly reduced Mago binding, suggesting that splicing can contribute to Mago enrichment at promoters (Supplementary Figure 6g). Finally, pre-treatment with a general inhibitor of RNA Pol II such as α-amanitin reduced Mago binding at TSS (Supplementary Figure 6h). A similar result was obtained when Pol II initiation was blocked using Triptolide treatment, while preventing Pol II elongation after pausing using DRB did not alter Mago binding (Supplementary Figure 6i, j). Collectively our data suggest that pre-EJC components bind to promoters via Ser5P Pol II, and that nascent RNA is required to stabilize this interaction.

**Pre-EJC binding to nascent RNA increases Pol II pausing**. To define the relationship between pre-EJC-binding and promoter proximal pausing, we evaluated all genes bound by pre-EJC components ($n = 816$) by several criteria. First, heatmaps show a positive correlation between Pol II occupancy and pre-EJC binding at TSS (Fig. 2e). Consistently, the proportion of Mago or pre-EJC-bound genes was higher at highly expressed genes (Fig. 2f and Supplementary Figure 6k). Second, we noticed that Mago was highly enriched at the TSS of strongly paused genes, which have a low PRR (Supplementary Figure 6l). Lastly, we found a positive correlation between pre-EJC binding and changes in Ser2P levels upon Mago KD (Fig. 2g, $p < 2.2 \times 10^{-16}$). Altogether these results suggest that pre-EJC binding to promoters might modulate promoter-proximal pausing.

To examine whether the pre-EJC is sufficient to promote Pol II pausing, we tethered Mago to the 5' end of a nascent RNA via the λN-boxB heterologous system[59]. Compared to λN alone, ectopic expression of λN-Mago ($p = 9.5 \times 10^{-5}$) led to increased enrichment of Pol II at the luciferase promoter and a slight depletion of

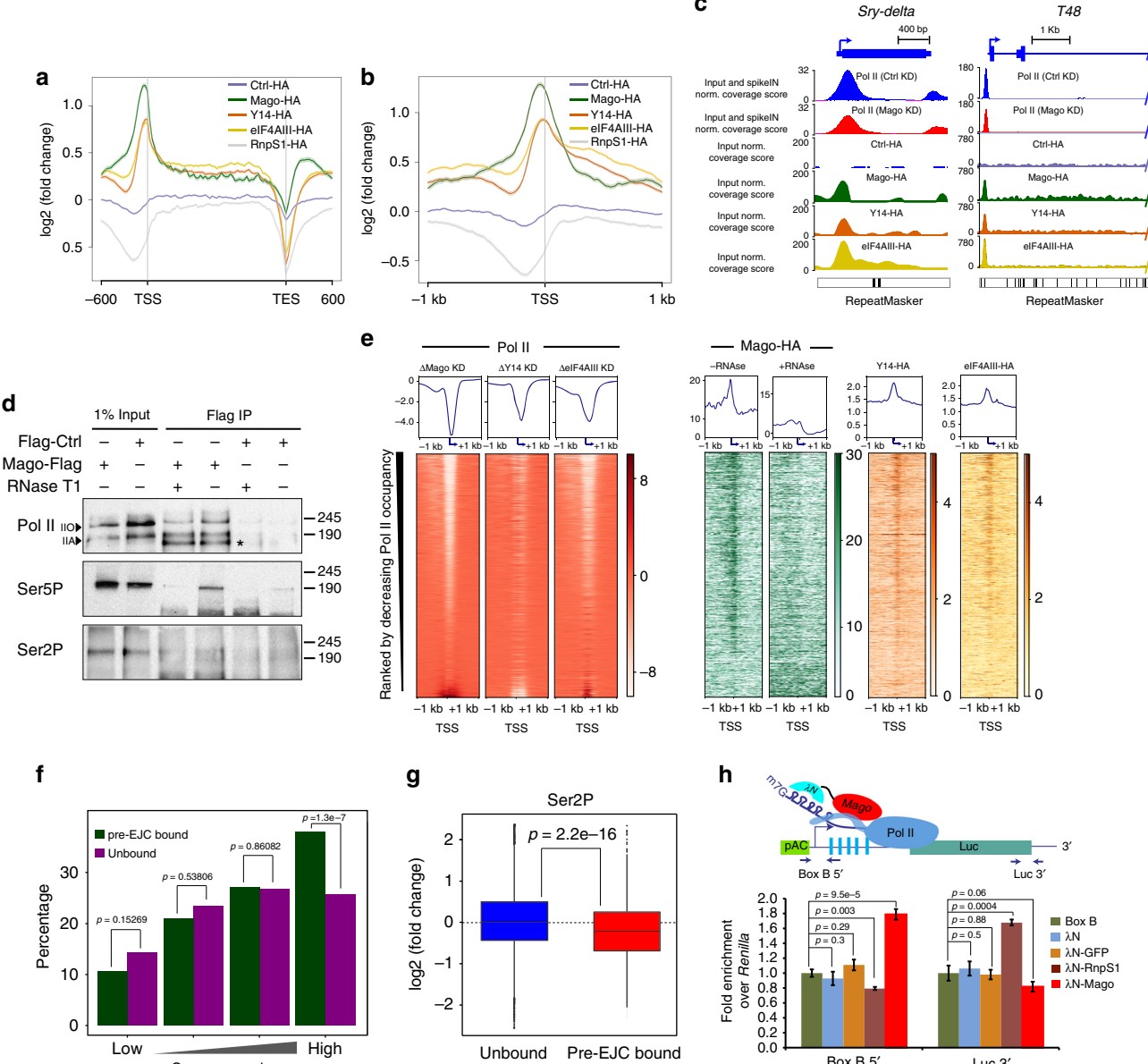

**Fig. 2** Mago binding to promoter regions modulates Pol II pausing. **a** Metagene profiles of ChIP-Seq performed with HA-tagged Mago, Y14, eIF4AIII, and Ctrl, with standard error of the mean for all the expressed genes based on averaged enrichment over input for two independent biological replicates. Log2 fold changes against input control are shown on *Y*-axis while *X*-axis depicts scaled genomic coordinates. **b** Metagene profiles with standard error of mean based on average enrichment over input for two independent biological replicates of ChIP-Seq performed with HA-tagged Mago, Y14, eIF4AIII, and Ctrl, centered at the TSS in a ±1 Kb window. Log2 fold changes against input control are shown on *Y*-axis while *X*-axis depicts genomic coordinates. **c** Input normalized and replicate averaged track examples of ChIP-Seq experiments from S2R+ cell extracts transfected with HA-tagged Mago, Y14, eIF4AIII, or Ctrl. Shown here is recruitment of pre-EJC components to an intronless (*Sry-delta*) and intron-containing (*T48*) genes. **d** Co-immunoprecipitation using anti-Flag antibody from cell extracts expressing either Flag-Mago or Flag alone (Flag-Ctrl), revealed with total Pol II, Ser5P and Ser2P antibodies. Note that Mago interacts with total Pol II, both hypo (IIA) and hyper phosphorylated (IIO) forms (indicated with the arrowheads). Mago also interacts with Ser5P but not with Ser2P, and this interaction with Pol II is partially dependent on RNA. **e** Heatmaps of HA-tagged pre-EJC components and change in total Pol II, centered at the TSS (−1 kb to +1 kb). Rows indicate all the genes bound by Pol II and are sorted by decreasing Pol II occupancy. The color labels to the right indicate the levels of enrichment. **f** Histogram showing percentage of pre-EJC bound genes amongst different quartiles of genes expressed in control condition. For quartile classification, all of the expressed genes in S2R+ cells were divided into four equal sized quartiles according to the level of expression, from low to high level. *p*-values for significance of the associations, derived from Fisher's exact test, are shown on top of the histogram. **g** Log2 fold changes in Ser2P level at the TSS upon control and Mago knockdowns, when separated according to pre-EJC binding. *p*-value is derived from a two-sample *t*-test. **h** Recruitment of Mago at the 5′ end of RNA is sufficient to induce pausing. (Top) Schematic of the BoxB-λN tethering assay. BoxB sequences (blue rectangles) were inserted upstream of the CDS of Firefly luciferase (green rectangle). The λN peptide (blue) was fused to Mago (shown in red), GFP or RnpS1, and transfected into S2R+ cells along with the modified Firefly luciferase plasmid as well as with a *Renilla* luciferase construct. (Bottom) Quantification of the ChIP experiment. Chromatin was prepared for the different conditions and followed by immunoprecipitation using antibody directed against total Pol II. The enrichment of Pol II at the promoter and at the 3′ end of Firefly luciferase was calculated after normalizing against a negative loci and *Renilla*. The enrichment for three independent biological replicates is shown along with *p*-values for tested conditions

Pol II at the 3′ end of luciferase (Fig. 2h). In contrast, ectopic expression of λN-GFP had no effect, whereas expression of λN-RnpS1 had an opposite effect regarding Pol II occupancy (Fig. 2h). These tethering experiments were repeated using additional genes selected on specific criteria: *piwi*, whose splicing is dependent on pre-EJC[43,44]; BBS8, which is unbound by the pre-EJC but has a paused Pol II; Crk, which is unbound by the pre-EJC and does not have a paused Pol II. In all conditions, tethering Mago to their 5′ UTR increased Pol II occupancy at 5′ end of the corresponding locus. Moreover, the decrease in Pol II occupancy observed upon Mago KD could be rescued by tethering Mago but not the GFP control (Supplementary Figure 7a). Thus, Mago recruitment to the 5′ end of nascent RNA is sufficient to increase promoter-proximal pausing of RNA Pol II at the corresponding locus, irrespective of whether the endogenous gene is bound by the pre-EJC.

**Loss of Mago results in changes in chromatin accessibility**. Transcription is tightly coupled to chromatin architecture[60]. To address whether KD of pre-EJC components affects chromatin organization, we performed MNase-Seq. We observed an increase in nucleosomal occupancy at the TSS upon depletion of Mago (Fig. 3a, b), consistent with elevated promoter-proximal pausing and with previous reports that paused Pol II competes with nucleosomes at TSS[60]. Furthermore, the phasing of nucleosomes within the gene body was strongly altered upon Mago depletion (Fig. 3a, b). Pre-EJC-bound promoters showed the most significant changes, consistent with a direct effect (Fig. 3c, $p < 2.2 \times 10^{-16}$). We also detected a mild but significant negative correlation between changes in Ser2P enrichment and chromatin accessibility (Fig. 3d, coefficient of determination $R^2 = -0.2274$). Mago KD also led to depletion of the activating histone mark H3K4me3, in particular at pre-EJC-bound genes (Fig. 3e, $p < 2.2 \times 10^{-16}$). Therefore, Mago modulates histone marks and chromatin accessibility, likely via its promoter-proximal pausing activity.

**Pre-EJC gene size dependency is mediated transcriptionally**. Depletion of pre-EJC components primarily affected the expression of genes containing larger introns (Supplementary Figure 1). We hypothesized that the underlying transcriptional changes upon Mago depletion might drive this size dependency. Indeed, Mago depletion led to an intron-size dependent increase in nucleosomal occupancy at promoters, and decrease in nucleosome occupancy along the gene body and at the TES (Fig. 4a). In contrast, Ser2P enrichment displayed anti-correlative changes with respect to the nucleosomal occupancy (Fig. 4b). Further, the increase in PRR upon Mago depletion also correlated with intron size (Fig. 4c). Thus, Mago has a stronger impact on the transcriptional regulation of genes with longer introns than genes with shorter introns. To determine whether these changes in nucleosomal and Ser2P occupancies result from pre-EJC binding, we calculated the percentage of genes bound by the pre-EJC in different classes relative to their representation in the total number of expressed genes. Interestingly, we found that pre-EJC binding was significantly over represented at genes containing longer introns (Fig. 4d, $p < 2.2 \times 10^{-16}$). Consistent with a direct control of gene expression by pre-EJC components on long intron-containing genes, we found that expression of pre-EJC-bound genes was significantly decreased upon KD of pre-EJC components (Fig. 4e–g, $p < 2.2 \times 10^{-16}$) and this decrease was also largely observed at nascent RNA (Fig. 4h, $p < 2.2 \times 10^{-16}$). Collectively, our results suggest that pre-EJC components preferentially bind and regulate the expression of large intron-containing genes via a direct transcriptional effect.

**Mago restricts P-TEFb binding to Pol II**. The P-TEFb complex induces Pol II release by promoting NELF and Ser2 phosphorylation of Pol II. To determine whether pre-EJC components influence pausing through an interplay with NELF we re-analyzed the publicly dataset available from the ref. [60]. We first noticed that NELF binds substantially more genes in comparison to the pre-EJC (3796 vs. 816), and that 45% of pre-EJC-bound genes does not overlap with NELF binding (Supplementary Figure 8a). Furthermore, like Mago, NELF-bound genes are over-represented on highly expressed genes (Supplementary Fig. 8b, c) and on promoters that are strongly paused (Supplementary Figure 8d, e). Accordingly, NELF KD affects PRR more strongly on highly paused genes (Supplementary Figure 8f, g). Given the similarity of NELF and Mago on pausing we tested whether their binding to promoters was dependent on each other. However, we found only minor effect on their binding upon the respective KD (Supplementary Figure 8h, j). Furthermore, NELF does not bind *MAPK* and its depletion had no effect on *MAPK* splicing (Supplementary Figure 8k), strongly suggesting independent mode of actions.

To determine whether the pre-EJC influences pausing by regulating P-TEFb occupancy, we monitored occupancy of Cdk9 via DamID[61,62]. We expressed N-terminally tagged Dam-Cdk9 in control and Mago-depleted S2R+ cells and observed increased Cdk9 enrichment at the TSS upon Mago depletion (Fig. 5a, b). Furthermore, the increase in Cdk9 enrichment correlated with Pol II occupancy (Fig. 5c). To validate the increased occupancy of Cdk9 in the absence of Mago we also performed ChIP-qPCR. In agreement with the DamID result, Cdk9 occupancy was increased at the 5′ end of *MAPK* (Fig. 5d). Importantly, Mago KD did not alter Cdk9 levels, indicating that this increased enrichment was not due to changes in protein expression (Supplementary Figure 9a). To evaluate whether the change in Cdk9 occupancy was directly driven by Mago occupancy, we analyzed pre-EJC-bound and unbound genes. The increase in Cdk9 enrichment for pre-EJC-bound class was mild albeit significantly higher than the unbound class (Fig. 5e, $p = 0.02508$). These data suggest that pre-EJC binding at the TSS controls Ser2 phosphorylation and Pol II pausing by restricting Cdk9 recruitment.

To address whether Mago inhibits P-TEFb recruitment by restricting its binding to Pol II, we evaluated the association of Cdk9 with Ser5P Pol II. We immunoprecipitated HA-SBP-tagged Cdk9 from control and pre-EJC KD cells and observed a substantial increase in the interaction between Cdk9 and Ser5P Pol II upon Mago depletion (Fig. 5f). Similar results were obtained upon KD of other pre-EJC components. Thus, these data strongly suggest that the pre-EJC restricts binding of P-TEFb to Pol II, which in turn reduces Ser2P levels and the entry of Pol II into elongation.

**Reducing Pol II release rescues Mago defects in vivo**. We hypothesized that reduced Pol II pausing upon Mago KD accounts for some of the increased exon skipping. To test this hypothesis, we attempted to rescue the splicing defects by decreasing the release of Pol II into gene bodies via simultaneously depleting Cdk9. We found that Cdk9 KD restored Ser2P levels upon Mago depletion (Fig. 6a) and partially rescued Ser2P occupancy at the *MAPK* gene (Fig. 6b). Further, the dependence of gene expression on intron size observed upon Mago KD was lost in the double KD (Supplementary Figure 9b). These data suggest that Cdk9 and Mago antagonistically regulate transcription. Importantly, reducing Cdk9 levels almost fully rescued *MAPK* splicing (Fig. 6c–e) as well as other Mago-dependent exon skipping events (Fig. 6f, $p = 8 \times 10^{-8}$) in Mago KD cells. Consistent with the pre-EJC influencing splicing via modulation of

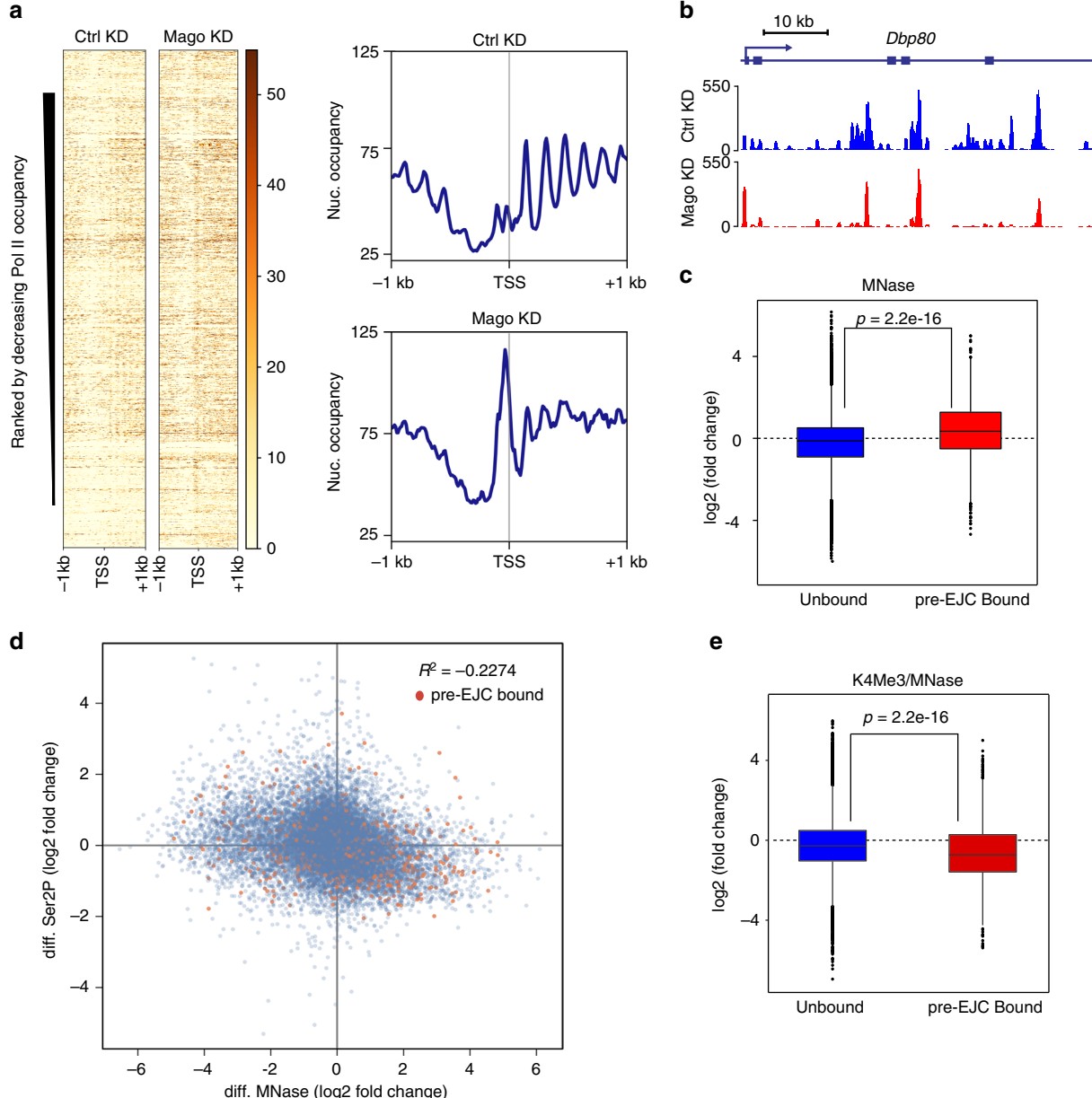

**Fig. 3** Mago controls chromatin accessibility. **a** Heatmaps of nucleosomal occupancy from MNase-seq experiments performed in biological duplicates from S2R+ cells in control or Mago knockdown conditions centered at the TSS in a ±1 Kb window. Rows indicate all the genes bound by Pol II and are sorted by decreasing Pol II occupancy. The color labels to the right indicate the levels of nucleosomal occupancy. Composite metagene profiles are also shown, with nucleosomal occupancy level on *Y*-axis and genomic coordinates on *X*-axis. **b** Genome browser view of averaged enrichment MNase-seq data from two independent biological duplicates in S2R+ cells treated with control or Mago double stranded RNA. The example shown here is *Dbp80* gene, a well-defined pre-EJC target. **c** Log2 fold changes in nucleosomal occupancy at the TSS (250 bp upstream and downstream of TSS) after control and Mago knockdowns. The changes were separated according to pre-EJC binding and a two-sample *t*-test was performed. **d** Scatterplot between changes in nucleosome and Ser2P occupancy after either control or Mago knockdown. Pre-EJC-bound promoters are highlighted by orange color. A mild negative correlation, as shown in the indicated pearson coefficient of correlation, between nucleosome and Ser2P occupancies was found. **e** Log2 fold changes in K4Me3 levels normalized to the nucleosomal occupancy (MNase data) at the TSS (250 bp upstream and downstream of TSS), after control and Mago knockdowns. The changes were separated according to pre-EJC binding and a two-sample *t*-test was performed

promoter-proximal pausing, we found that genes that display differential splicing upon depletion of pre-EJC components were significantly enriched for pre-EJC binding (Supplementary Figure 9c–e, Fisher's test $p < 2.2 \times 10^{-16}$). Furthermore, the tethering of Mago to the 5′ end of piwi or Crk that increased Pol II pausing was sufficient to rescue splicing defects associated with the Mago KD (Supplementary Figure 7b). Lastly, depletion of Cdk12, another kinase involved in the release of promoter-proximal pausing[26], also rescued *MAPK* splicing of Mago-

depleted cells (Supplementary Figure 9f, g). Altogether, these results strongly suggest that Mago regulates gene expression and exon definition via regulation of Pol II promoter-proximal pausing.

*MAPK* is the main target of the EJC during *Drosophila* eye development[46]. As shown previously, eye-specific depletion of Mago strongly impairs photoreceptor differentiation[45,46]. Strikingly, decreasing Cdk9 function in a similar background rescued eye development (Fig. 6g). Notably, the number of differentiated

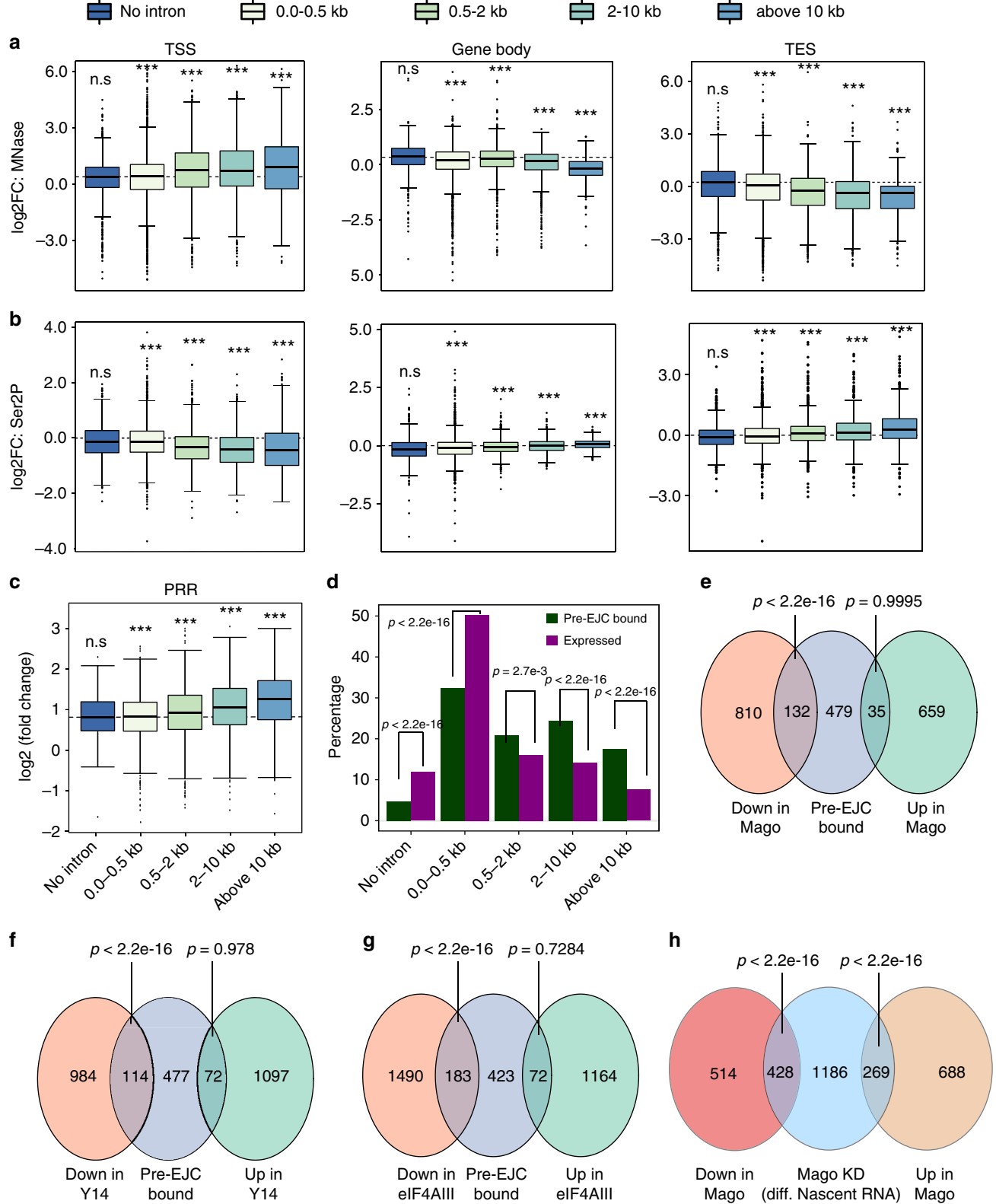

photoreceptors in larvae and adults was substantially increased. We observed a similar rescue of photoreceptor differentiation in a double KD for Mago and Cdk12 (Supplementary Figure 10a). Further, depletion of Cdk9 rescued the lethality and the eye defects associated with eIF4AIII KD (Supplementary Figure 10b). In contrast, reducing the speed of Pol II or depleting several transcription elongation factors failed to substantially rescue eye

development in the absence of Mago (Supplementary Figure 10c–f), providing additional evidence that Mago transcriptional function occurs at the level of promoter-proximal pausing rather than at the transcription elongation stage. Thus, despite the numerous post-transcriptional functions of the EJC, modulating Pol II release is sufficient to rescue the eye defect associated with pre-EJC depletion.

**Fig. 4** Mago controls chromatin accessibility and Pol II pausing in a gene size dependent manner. **a**, **b** Changes in nucleosome occupancy (**a**) and Ser2P levels (**b**) in control and Mago knockdown in S2R+ cells at the promoter, gene body, and TES, separated according to the size of the largest intron (ANOVA, $p < 2.2 \times 10^{-16}$). **c** Box plots showing changes in PRRs upon Mago depletion when compared to control, separated into different intron classes. **d** Percentage of pre-EJC-bound genes in different intron classes, along with the percentage of each class amongst all the expressed genes. The proportion of pre-EJC-bound genes is increased for genes containing larger introns, relative to their abundance. *p*-values are derived from Fisher's exact test. **e–g** Venn diagrams showing the overlap between Mago-bound, Y14-bound, and eIF4A3-bound genes and genes with differential expression upon respective knockdowns. The overlap between genes that are downregulated upon each pre-EJC component knockdown, and their respective target genes is significant. *p*-values are derived from Fisher's exact test. **h** Venn diagram showing the overlap between genes identified as differentially expressed in mRNA sequencing and nascent RNA sequencing (4sU-Seq) upon Mago KD. *p*-values are derived from Fisher's exact test, separated for either upregulated or downregulated genes

**The function of Mago in Pol II pausing is conserved**. To determine if EJC-mediated promoter-proximal pausing is conserved in vertebrates, we investigated the function of Magoh, the human ortholog of *Drosophila* Mago. We found that depletion of Magoh in HeLa cells led to an increased release of Pol II from the promoter to the gene body, and in turn to a higher PRR (Fig. 7a–d), as well as higher level of Ser2P, but not of Ser5P (Fig. 7e). Additionally, Magoh specifically interacted with Pol II and Ser5P, but not Ser2P (Fig. 7f–h). Finally, we immunoprecipitated Cdk9 from control and Magoh KD cells, and observed a stronger interaction between Ser5P and Cdk9 upon depletion of Magoh (Fig. 7i). Thus, the function and mechanism of the pre-EJC in the control of promoter proximal pausing is conserved in human cells.

## Discussion

Our work uncovers an unexpected connection between the nuclear EJC and the transcription machinery via the regulation of Pol II pausing, which is conserved from flies to human. The pre-EJC stabilizes Pol II in a paused state, at least in part, by restricting the association of P-TEFb with Pol II via non-canonical binding to promoter regions. The premature release of Pol II into elongation in absence of the EJC results in splicing defects, highlighting the importance of this regulatory step in controlling downstream RNA processing events (Fig. 7j).

Promoter proximal pausing is a widespread transcriptional checkpoint, whose functions and mechanisms have been extensively studied. Several regulators have been identified, which includes P-TEFb, NELF and DSIF. Our data reveal that the pre-EJC plays a similar role as the previously described negative factors by preventing premature Pol II release into elongation. How does the pre-EJC control Pol II pausing and how does it interplay with other pausing regulators? Our study provides some answers to these questions. In absence of pre-EJC components, P-TEFb associates more strongly with Pol II, which results in increased Ser2 phosphorylation, demonstrating that one of the activities of the pre-EJC is to restrain P-TEFb function by diminishing its association with chromatin. While it is not clear yet how the pre-EJC exerts this function, a simple mechanism would be by steric interference for Pol II binding, although more indirect mechanisms might also exist. This mechanism infers that both the pre-EJC and Cdk9 bind similar sites on the CTD on Pol II, which fits with the association of the pre-EJC with the Ser5 phosphorylated form of Pol II and not with Ser2P. However, we also found elevated Cdk9 binding and premature release of Pol II at Mago-unbound genes, albeit to a lesser extent compared to Mago-bound genes, suggesting that additional mechanisms must be involved.

It is interesting to note that the binding of the pre-EJC to Pol II requires the presence of nascent RNA. A recent study also supports these findings showing specific association of pre-EJC components on polytene chromosomes that depends on nascent

transcription but is independent of splicing[38]. This is reminiscent to the binding of DSIF and NELF[63–68], suggesting that interaction with Pol II and stabilization via nascent RNA is a general mechanism to ensure that pausing regulators exert their function at the right time and at the right location. Upon external cues, P-TEFb modifies the activities of both NELF and DSIF through phosphorylation, promoting Pol II release. It would be of interest to address whether P-TEFb also regulates the EJC in a similar manner. Intriguingly, previous studies revealed that eIF4AIII is present in the nuclear cap-binding complex[69], while Y14 directly recognizes and binds the mRNA cap structure[70,71]. It is therefore possible that this cap-binding activity confers the ability of the EJC to bind nascent RNA. Consistent with this hypothesis, the KD of Cap binding protein (Cbp) strongly reduced association of Mago to chromatin (Supplementary Figure 8h). Nevertheless, since Cbp is also required to stabilize transcripts, the reduced Mago binding might result from this confounding effect. Moreover, other factors must be clearly involved as only a subset of genes is bound by the pre-EJC.

SRSF2 is another splicing regulator that was previously demonstrated to modulate Pol II pausing via binding to nascent RNAs[72]. In this case, SRSF2 exerts an opposite effect by facilitating Pol II release into the elongation phase. This effect occurs via increased P-TEFb recruitment to gene promoters. Although we have not found convincing evidence for a conserved role of the *Drosophila* SRSF2 homolog in this process (unpublished data), one may envision that the pre-EJC counteracts the effect of SRSF2 to stabilize Pol II pausing. Consistent with this possibility, EJC binding sites are often associated with RNA motifs that resemble the binding sites for SR proteins[73]. It is therefore possible that SR proteins influence pre-EJC loading to mRNA and vice versa.

Our previous work along with studies from other groups suggested that the EJC modulates splicing by two distinct mechanisms[43–46]. On one hand, the EJC facilitates the recognition and removal of weak introns after prior deposition to flanking exon-exon boundaries. We proposed that EJC deposition occurs in a splicing dependent manner after rapid removal of bona fide introns, which are present in the same transcript. Thus, a mixture of "strong" and "weak" introns ensures EJC's requirement in helping intron definition. This function requires the activity of the EJC splicing subunits Acinus and RnpS1, which are likely involved in the subsequent recruitment of the splicing machinery near the weak introns. While this model is attractive, it does not however explain every EJC-regulated splicing event. In particular, depletion of pre-EJC components results in a myriad of exon-skipping events, which occur frequently on large intron-containing transcripts (this study and refs. [45,46]). In contrast to intron definition, this exon definition activity only slightly required the EJC splicing subunits, suggesting an additional mechanism. We now show that the pre-EJC controls exon definition at least in part by preventing premature release of Pol II into transcription elongation. Our results shed light on a recent

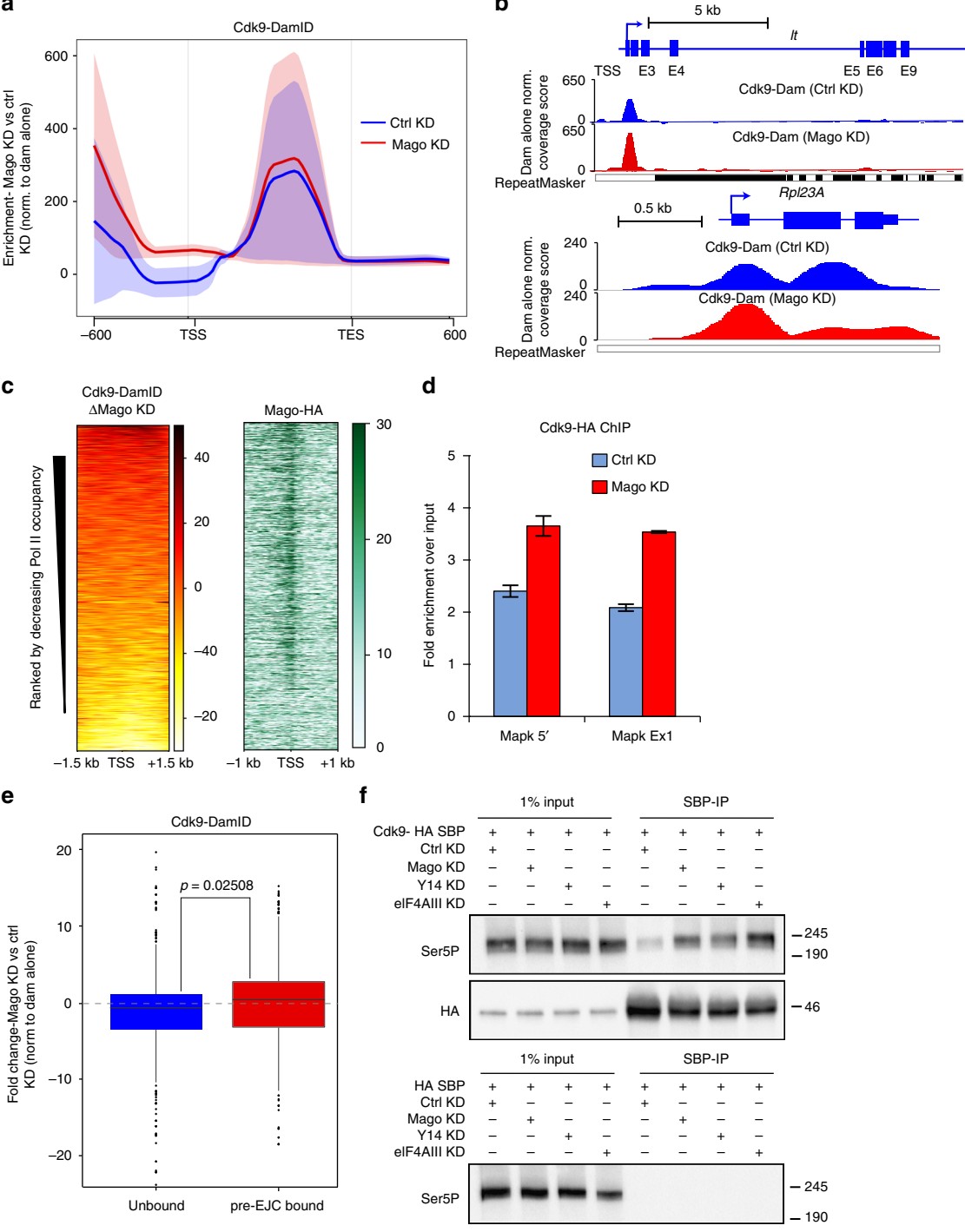

**Fig. 5** Loss of pre-EJC core components results in increased Cdk9 binding to Pol II. **a** Metagene profiles of Cdk9-DamID in control and Mago knockdown cells as averaged enrichment over input from two independent biological replicates with standard error of the mean for all the expressed genes. Fold enrichment represented on *Y*-axis was calculated against Dam alone control in respective conditions (using damidseq_pipeline), while *X*-axis depicts scaled genomic coordinates. **b** Genome browser view of Dam alone normalized and averaged tracks of Cdk9-DamID for *light* and *Rpl23A*. **c** Heatmaps of changes in normalized Cdk9-DamID enrichment in Mago depleted S2R+ cells compared to control, centered at the TSS in a ±1.5 Kb window. Rows indicate all the genes bound by Pol II and are sorted by decreasing Pol II occupancy, and the color labels to the right indicate the level of enrichment. Heatmap of Mago-HA centered at the TSS (−1 kb to +1 kb) is also shown. **d** ChIP-qPCR analysis of Cdk9-HA occupancies at indicated *MAPK* locus. Error bars indicate the standard deviation of three biological replicates. **e** Changes in Cdk9 occupancy at the TSS (250 bp upstream and downstream of TSS) after control and Mago knockdowns, calculated from Cdk9-DamID experiment, after normalizing to the Dam alone control. The changes were separated according to pre-EJC binding and a two-sample *t*-test was performed. **f** Co-immunoprecipitation experiments from S2R+ cells extracts expressing either HA-SBP-Cdk9 or HA-SBP alone, revealed with Pol II Ser5P antibody. Immunoprecipitations were performed from control cells or cells depleted for pre-EJC core components (Mago, Y14, and eIF4AIII). Shown also is the western blot against HA tag to assess the efficiency of the pull-down of HA-SBP-Cdk9 in each knockdown condition

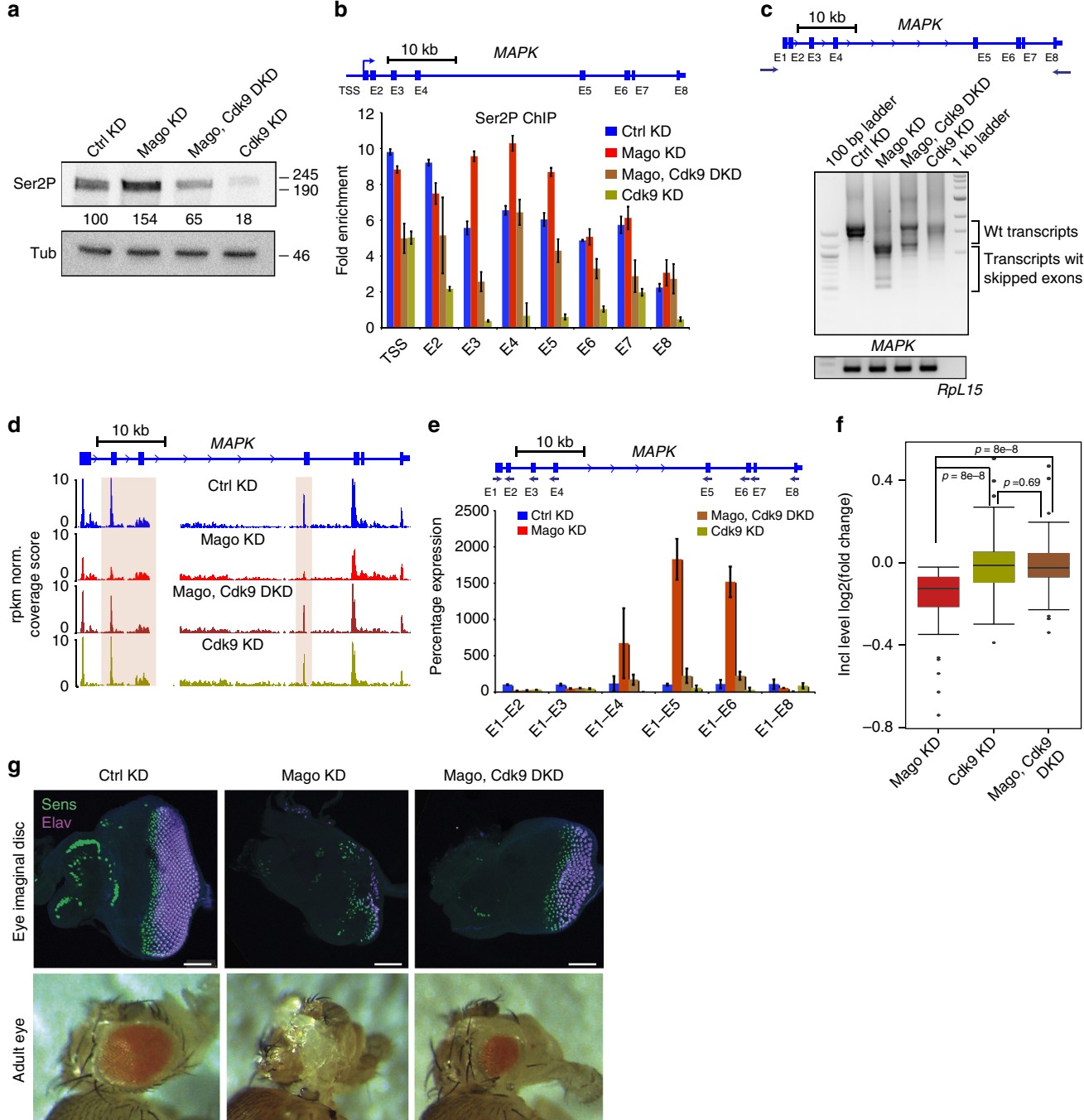

**Fig. 6** Restoring pausing is sufficient to rescue Mago-associated exon skipping defects. **a** Mago knockdown results in elevated level of Ser2P phosphorylation of Pol II. Western blots using antibodies against Pol II Ser2P and Tubulin, using S2R+ cell extracts with indicated knockdowns. Signal in the knockdown conditions was normalized to the control condition using Tubulin as loading control and quantification of the intensity was performed with ImageJ. **b** ChIP-qPCR analysis of Ser2P occupancy level at *MAPK* locus in the indicated knockdowns. The primers used for the analysis are indicated in the scheme above. Bars indicate the 95% confidence interval from the mean of two biological replicates. **c** Agarose gels showing RT-PCR products for *MAPK* and *RpL15* using RNA extracted from S2R+ cells with indicated knockdowns as template for cDNA synthesis. The primers used for the PCR 5′ and 3′ UTR of *MAPK* are shown in the scheme above. RT-PCR products for *RpL15* from respective knockdown condition were used as loading control. **d** Replicate averaged RNA-Seq track examples of *MAPK* in several knockdown conditions. Mago KD results in several exon skipping events, which are rescued upon simultaneous knockdown of Cdk9. The exons skipped in Mago knockdown condition are highlighted by colored rectangles **e** Quantitative RT-PCR using RNA extracts derived from S2R+ cells with the indicated knockdowns. The amplicons were obtained using the same 5′ forward primer (E1) together with the reverse primers on respective exons, as shown in the scheme above. The level of exon skipping is compared to the control treatment, with *RpL49* used for normalization. Bars indicate the 95% confidence interval from the mean of two biological replicates. **f** Box plots showing the log2 fold change in inclusion level of alternatively spliced exons in the indicated knockdowns (rMATS was used for the analysis). **g** Upper panel: Staining of eye imaginal discs from third instar larvae with indicated dsRNAs specifically expressed in the eye (using the *ey*-GAL4 driver). All photoreceptors are stained with anti-Elav antibody (purple), and R8 (the first class of photoreceptor to be specified) with anti-Senseless (green). Scale bar 50 μM. Lower panel: Adult eyes of a control fly and flies with indicated KD

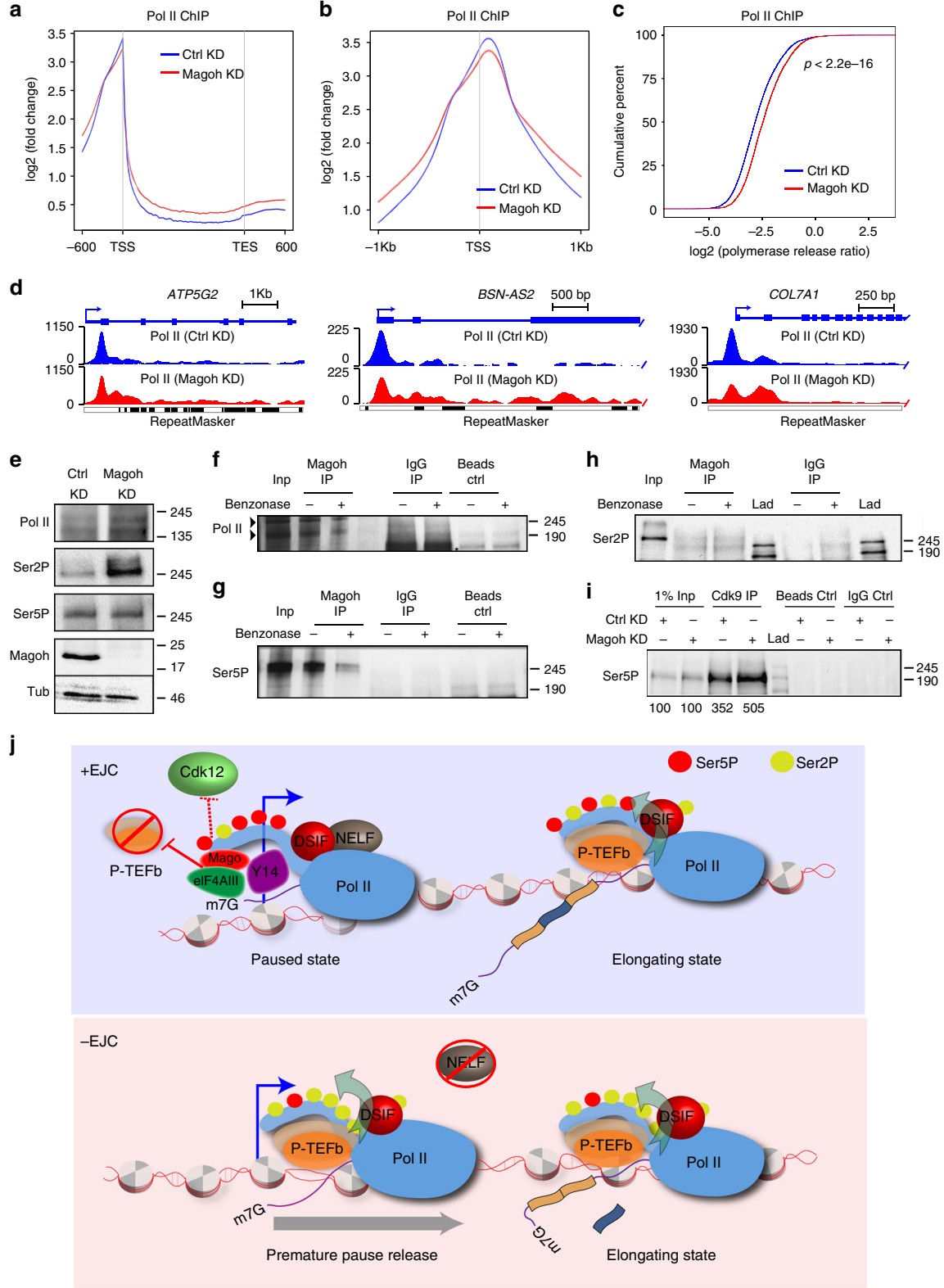

observation in human cells showing that the usage of general transcription inhibitors improve splicing efficiency on two EJC-mediated exon skipping events[47]. Of note, two recent studies suggested a third mechanism for splicing regulation by the EJC that involves the repression of cryptic splice sites (PMID: 30388410, 30388411).

The notion that splicing takes place co-transcriptionally is now a general consensus and two non-exclusive models regarding the

impact of transcription on splicing have been proposed. Through the ability of the C-terminal repeat domain (CTD) of its large subunit, RNA Pol II can recruit a wide range of proteins to nascent transcripts[74–77], thereby influencing intron removal. Pol II can influence splicing via a second mechanism referred to as kinetic coupling. According to the model, changes of elongation rates can alter the recognition of exons containing weak splice sites[78–80]. In regards to pre-EJC's activity we favor the first model.

**Fig. 7** The role of Mago on promoter proximal pausing is conserved in human. **a** Metagene profiles based on averaged enrichment over input from two independent biological replicates of total Pol II occupancies in control and Magoh-depleted HeLa cells with standard error of the mean for all the Pol II bound genes. Log2 fold changes against input control are shown on Y-axis while X-axis depicts scaled genomic coordinates. **b** Metagene profiles of averaged enrichment from two independent biological replicates of total Pol II occupancies in control and Magoh-depleted HeLa cells with standard error of mean for all the Pol II bound genes, centered at the TSS in a ±1 Kb window. Log2 fold changes against input control are shown on Y-axis while X-axis depicts genomic coordinates. **c** The ECDF plot of PRR in HeLa cells treated with either control or *Magoh* siRNA. *p*-value is derived from two-sample Kolmogorov–Smirnov test. **d** Track examples of total Pol II ChIP-Seq from HeLa cells extracts, after either control or Magoh knockdown. The tracks are the average of two independent biological replicates after input normalization. **e** Western blots performed using antibodies against total Pol II, Ser2P, Ser5P, and Magoh, in HeLa cells treated with either control or *Magoh* siRNA. Similar to *Drosophila*, the loss of Magoh leads to elevated level of Ser2P without affecting Ser5P. **f–h** Co-immunoprecipitation of Magoh from HeLa cells extracts, using antibody directed against Magoh. Western blots using Pol II, Ser5P and Ser2P antibody reveal RNA-dependent interaction of Magoh with Pol II and Ser5P. There was no detectable interaction of Magoh with Ser2P. The specific bands for Pol II are highlighted by arrowheads in **f**. The lane labeled with "Lad" indicates ladder. **i** Co-immunoprecipitation of Cdk9 using anti-Cdk9 antibody from HeLa cells extracts treated with either control or *Magoh* siRNA. Western blot was performed with anti-Ser5P antibody. The quantification of the intensity was performed with ImageJ. **j** Model: The pre-EJC stabilizes Pol II pausing by restricting P-TEFb binding at promoter, and possibly by sequestering Cdk12. This activity is required for proper recognition of exons

First, our genome wide studies demonstrate a global impact of the pre-EJC on promoter proximal pausing. Second, we did not observe substantial alteration of the average rate of transcription elongation upon Mago depletion. We did find however some gene-to-gene differences but they poorly correlate with the degree of exon inclusion. Still, this effect might be a secondary consequence of splicing defects, as a previous study suggested the existence of a splicing-dependent elongation checkpoint[81]. Third, reducing the speed of Pol II or depleting the function of transcription elongation factors failed to rescue Mago-splicing defects, arguing that the positive impact of reducing P-TEFb levels on exon definition is dependent on its function in Pol II release rather than in regulating the elongation stage. Thus, in the light of previous model regarding the interplay between pre-mRNA capping and transcription, we propose that by stabilizing Pol II pausing the EJC provides enough time for the recruitment of additional splicing factors that play a critical role in exon definition.

Pol II pausing is found more prominently at developmentally regulated genes, which tend to be long and frequently regulated by alternative splicing. We found a size dependency for Mago-bound genes as well as for Mago-regulated gene expression, suggesting that pre-EJC function is adapted to regulate exon definition of large genes by enhancing their promoter proximal pausing. Interestingly, a recent study shows that genes with long introns tend to be spliced faster and more accurately[82]. Whether this function depends on EJC binding to nascent RNA constitutes an interesting possibility. The next important challenge will be to address the precise mechanisms by which promoter proximal pausing influences pre-mRNA splicing at these developmental genes.

## Methods
**Cloning**. The plasmids used for chromatin immunoprecipitation (ChIP) and co-IP assays in *Drosophila* S2R+ cells were constructed by cloning the corresponding cDNA in the pPAC vector either with N-terminal Flag—3× HA tag or with HA-SBP tag. The CDS were cloned in pPAC vector with N-terminal tags between EcoRV and NotI. The lambdaN and Box-B constructs are derived from the plasmids described earlier[83]. The lambdaN constructs were made by cloning different CDS in frame at the C-terminal, between EcoRV and NotI sites. The boxB constructs were made by cutting out the 3′ boxB sites, and cloning it upstream of luciferase gene with KpnI site. For endogenous genes, the luciferase CDS was first removed using SpeI, followed by blunting the ends. BoxB sites were reintroduced using NotI and StuI sites, and these sites were used to clone endogenous genes.

**RNA isolation and RT-PCR**. RNA was extracted from cells using Trizol reagent, following the manufacturer's protocol. For reverse transcription, cDNA was synthesized using MMLV reverse transcriptase (Promega, Cat No-M1701). For semi-quantitative RT-PCR 2 μg of RNA was reverse transcribed. Five microliter of the cDNA was amplified using the respective primers in 50 μl PCR reaction, using One Taq polymerase (NEB, Cat No-M0480). After 40 cycles of amplification half of the

PCR product was loaded on 1% agarose gel to qualitatively analyze the splicing products. For real time PCR, RpL15 was used as an internal control. Relative abundance of transcripts was calculated by the $2^{\Delta Ct}$ method. PCR primers used for semi-quantitative and real time PCR are listed in Supplementary Table 1.

**Cell culture, RNAi, and transfection**. *Drosophila* S2R+ cells were cultured in Schneider Cell's Medium (GIBCO, Cat No-21720) supplemented with 10% FBS and 2% Penicillin/Streptomycin. The plasmids expressing various transgenes were transfected with Effectene transfection reagent (Qiagen, Cat No-301425), following manufacturer's protocol. For knock down experiments, dsRNA was synthesized overnight at 37 °C using Hi-Scribe T7 transcription kit (NEB, Cat No-E2040). dsRNA was transfected in S2R+ cells by serum starvation for 6 h. The treatment was repeated three times and cells were harvested 7 days after the first treatment for Mago. For knockdown of other pre-EJC components and Btz, the treatment was repeated two times and cells were harvested 5 days after the first treatment. The primers used for generating dsRNA are listed on Supplementary Table 1. S2R+ cells were treated with 50 μg/ml of α-amanitin for 7 h to block transcription. Triptolide treatment in S2R+ cells was performed at 10 μM for 6 h.

HeLa cells were cultured in standard RPMI medium supplemented with 10% FBS and 2% Penicillin/Streptomycin. For siRNA knockdown, cells were transfected with 10 nM of siRNA using RNAiMax (Invitrogen) according to manufacturer's protocol. Cells were harvested 48 h after transfection. A mixture of three siRNA was used to deplete Magoh, two against MagohA isoform (siRNA sequence; 1-CGGGAAGTTAAGATATGCCAA; 2-CAGGCTGTTTGTATATTTAAT) and one targeting MagohB isoform (siRNA sequence; GATATGCCAACAACAGCAA).

**Antibodies**. The following antibodies were used in this study. For total Pol II ChIP, RBP1 (Diagenode Cat No-15200004) was used. Ser2P ChIP was performed using ab5095 (Abcam); 3E10 (Chromotek) was used for western blotting. Anti-Ser5P Pol II (Chromotek, Cat No-3E8) and ARNA3 (Pol II) antibodies (Progen, Cat No-65123) were used for western blot assays. For immunoprecipitation experiments, anti-Flag M2 (Sigma, Cat No-F3165) and M-280 streptavidin beads (Thermo-Fisher, Cat No-11205) were used. A polyclonal rabbit anti-Mago antibody was generated from Metabion (Germany). Anti-hCdk9 (Santa Cruz, Cat No-8338) was used for immunoprecipitation from HeLa cells extracts. The Cdk9 western blot from S2R+ cell extract was performed by anti-dCdk9 antibody, a kind gift from Akira Nakamura.

**Immunostaining**. The primary antibodies used were rat anti-Elav (1:5; Developmental Studies Hybridoma Bank) and guinea pig anti-Senseless (1:1000)[84]. Eye imaginal discs were dissected in 0.1 M sodium phosphate buffer (pH 7.2) and then fixed in PEM (0.1 M PIPES at pH 7.2 mM MgSO4, 1 mM EGTA) containing 4% formaldehyde. Washes were done in 0.1 M phosphate buffer with 0.2% Triton X-100. Appropriate fluorescent-conjugated secondary antibodies were used (1:1000; Jackson Immunoresearch Laboratories). Images were collected on Zeiss TCS SP5 confocal microscope.

**Co-IP assay and western blot analysis**. For co-IP assay in S2R+ cells, cells were plated in 10 cm cell culture dish, and respective transgenes were transfected using Effectene transfection reagent, according to manufacturer's protocol. Forty-eight hours post transfection cells were collected, washed once with PBS and re-suspended in swelling buffer (10 mM Tris pH 7.5, 2 mM MgCl2, 5 mM MgCl2, 3 mM CaCl2, and protease inhibitors). After incubating 10 min on ice, the suspension was spun at 600 g for 10 min at 4 °C. After discarding the supernatant the pellet was resuspended in lysis buffer (10 mM Tris pH 7.5, 2 mM MgCl2, 5 mM MgCl2, 3 mM CaCl2, 0.5% NP-40, 10% glycerol and protease inhibitors) and centrifuged for 5 min at 600 × *g*. Nuclei were resuspended in lysis buffer (40 mM

HEPES pH7.4, 140 mM NaCl, 10 mM MgCl$_2$, 0.5% Triton X-100 and protease inhibitors) and sonicated in the bioruptor plus (Diagenode) for 6 cycles with 30 s "ON/OFF" at low settings. Protein concentrations were determined using Bradford reagent (BioRad, Cat No-5000006). For IP 2 mg of proteins were incubated with respective antibody in lysis buffer and rotated head-over-tail O/N at 4 °C. The beads were washed 3× for 10 min with lysis buffer and IP proteins were eluted by incubation in 1× SDS buffer at 85 °C for 10 min. Immunoprecipitated and input proteins were analyzed by western blot, after separating them on 4–15% gradient SDS-PAGE gel (BioRad, Cat No-4561083) and transferred to PVDF membrane (Millipore, Cat No-IPVH00010). After blocking with 5% milk in TBST (0,05% Tween in 10 mM Tris pH 7.4 and 140 mM NaCl) for O/N at 4 °C, the membrane was incubated with respective primary antibody in blocking solution O/N at 4 °C. The antibodies were used at following dilution: Ser2P: 1:500; Ser5P: 1:500; ARNA-3: 1:1000; HA: 1:2500; Mago: 1:2000; and Magoh: 1:1000. Membrane was washed 4× in TBST for 15 min and incubated 1 h at RT with secondary antibody in blocking solution. Blots were developed using SuperSignal™ West Pico Chemiluminescent Substrate (Thermo Scientific, Cat No-34080) and visualized using BioRad Gel documentation system. Full-length blots with molecular weight standards are provided in the Supplementary Figure 11.

For co-IP assay in HeLa, cells were plated in 10 cm cell culture dish. Afterwards, cells were collected washed once with PBS and re-suspended in swelling buffer (10 mM Tris pH 7.5, 2 mM MgCl$_2$, 5 mM MgCl$_2$, 3 mM CaCl$_2$, and protease inhibitors), identical to the approach as in S2R+ cells. After incubating 10 min on ice, the suspension was spun at 600 g for 10 min at 4 °C. After discarding the supernatant the pellet was resuspended in lysis buffer (10 mM Tris pH 7.5, 2 mM MgCl$_2$, 5 mM MgCl$_2$, 3 mM CaCl$_2$, 0.5% NP-40, 10% glycerol and protease inhibitors) and centrifuged for 5 min at 600 × g. Nuclei were resuspended in lysis buffer (40 mM HEPES pH7.4, 140 mM NaCl, 10 mM MgCl$_2$, 0.5% Triton X-100 and protease inhibitors) and sonicated in the bioruptor plus (Diagenode) for 6 cycles with 30 s "ON/OFF" at low settings. Protein concentrations were determined using Bradford reagent (BioRad, Cat No-5000006). For IP 2 mg of proteins were incubated with anti-Magoh antibody in lysis buffer and rotated head-over-tail O/N at 4 °C. The beads were washed 3× for 10 min with lysis buffer and IP proteins were eluted by incubation in 1× SDS buffer at 85 °C for 10 min, immunoprecipitated and input proteins were analyzed by western blot, as described above.

**RNA extraction and RNA-seq.** RNA was extracted from cells using Trizol reagent, following the manufacturer's protocol. RNA was further cleaned for organic contaminants by RNeasy MinElute Spin columns (Qiagen, Cat No-74204). The purified RNA was subjected to oligodT (NEB, Cat No-S1419S) selection to isolate mRNA. The resulting mRNA was fragmented and converted into libraries using illumina TruSeq Stranded mRNA Library Prep kit (illumina, Cat No- 20020594) following manufacturer's protocol.

**ChIP-qPCR and ChIP-Seq.** S2R+ cells and HeLa cells were fixed with 1% formaldehyde for 10 min at room temperature, and harvested in SDS buffer resuspended in RIPA buffer (140 mM NaCl, 10 mM Tris-HCl [pH 8.0], 1 mM EDTA, 1% Triton X-100, 0.1% SDS, 0.1% DOC), and lysed by sonication. The lysate was cleared by centrifugation, and incubated with respective antibodies overnight at 4 °C. Antibody complexes bound to protein G beads were washed once with 140 mM RIPA, four times with 500 mM RIPA, once with LiCl buffer and twice with TE buffer for 10 min each at 4 °C. DNA was recovered after reverse crosslinking and phenol chloroform extraction. After precipitating and pelleting, DNA was dissolved in 30 μl of TE. Control immunoprecipitations were done in parallel with either tag alone or knock down controls, and processed identically. Five microliters of immunoprecipitated DNA were used for checking enrichment with various primer pairs (listed in Supplementary Table 1) on Applied Biosystem ViiA™ 7 real time machine using SYBR green reagent (Life technologies, Cat No-4367659). To examine whether these changes in Pol II distribution were widespread, we performed ChIP-Seq experiments in control and Mago KD conditions. To exclude the possibility of changes in Pol II occupancy driven by differences in immunoprecipitation efficiency and technical variance during library preparation in different knock down conditions, we used yeast chromatin as "spike-in" control (Orlando et al.[85]). With this approach, we confirmed the decrease in Ser2P levels and Pol II at the promoter region and an increase within the gene body of MAPK. After validating enrichment, the recovered DNA was converted into libraries using NebNext Ultra DNA library preparation kit, following manufacturer's protocol. DNA libraries were multiplexed, pooled and sequenced on Illumina HiSeq 2000 platform.

**DRB-4sU-Seq.** S2R+ cells were grown in Schneider's Cell Medium with 10% bovine serum supplemented with antibiotics and maintained at 25 °C. 5,6-dichlorobenzimidazole 1-β-d-ribofuranoside (DRB) from Sigma (D1916) was used at a final concentration of 300 μM, dissolved in water, for 5 h. 4-thiouridine (4sU) was purchased from Sigma (Cat No-T4509) and used at a final concentration of 100 μM. Control and Mago KD was performed as described before. All the samples were labeled for 6 min with 4-thiouridine, and transcription was allowed to proceed after DRB removal for 0, 2, 8, and 16 min along with one non-DRB treated control.

A total of 100–130 μg RNA was used for the biotinylation reaction. 4sU-labeled RNA was biotinylated with EZ-Link Biotin-HPDP (Thermo Scientific, Cat No-21341), dissolved in dimethylformamide (DMF, Sigma Cat No-D4551) at a concentration of 1 mg/ml. Biotinylation was done in labeling buffer (10 mM Tris pH 7.4, 1 mM EDTA) and 0.2 mg/ml Biotin-HPDP for 2 h with rotation at room temperature. Two rounds of chloroform extractions removed unbound Biotin-HPDP. RNA was precipitated at 20,000×g for 20 min at 4 °C with a 1:10 volume of 5 M NaCl and an equal volume of isopropanol. The pellet was washed with 75% ethanol and precipitated again at 20,000×g for 10 min at 4 °C. The pellet was left to dry, followed by resuspension in 100 μl RNase-free water. Biotinylated RNA was captured using Dynabeads MyOne Streptavidin T1 beads (Invitrogen, Cat No-65601). Biotinylated RNA was incubated with 50 μl Dynabeads with rotation for 15 min at 25 °C. Beads were magnetically fixed and washed with 3× Dynabeads washing buffer. RNA-4sU was eluted with 100 μl of freshly prepared 100 mM dithiothreitol (DTT), and cleaned on RNeasy MinElute Spin columns (Qiagen, Cat No-74204). For the untreated 4sU-Seq version used for calculating polymerase release ratio (PRR), an identical approach was used with following modifications. During the period when biotinylated RNA was incubated with 50 μl Dynabeads with rotation for 15 min at 25 °C, RNAse T1 was added in order to fragment RNA to 100 bp. Beads were magnetically fixed and washed with 3× Dynabeads washing buffer, as described before. RNA-4sU was eluted with 100 μl of freshly prepared 100 mM DTT, and cleaned on RNeasy MinElute Spin columns (Qiagen, Cat No-74204). Enriched nascent RNAs were converted to cDNA libraries with Drosophila Ovation Kit (Nugen- Cat No-7102–32) with integrated ribosomal depletion workflow. Amplified cDNA libraries were pooled, multiplexed, and sequenced on two lanes of Illumina HiSeq 2000.

**MNase-Seq.** S2R+ cells were fixed with 1% formaldehyde for 10 min at RT. Cells were harvested and 20 million nuclei were spun at 3500 g at 4 °C for 10 min. Nuclear pellets were resuspended in 300 μl of MNase digestion buffer (0.5 mM spermidine, 0.075% NP40, 50 mM NaCl, 10 mM Tris-HCl, pH 7.5, 5 mM MgCl$_2$, 1 mM CaCl$_2$, 1 mM β-mercaptoethanol and complete protease inhibitors). Reaction was spun at 3200 g, 4 °C for 10 min and resuspended in 50 μl of MNase digestion buffer and digested with 30U of MNase at 37 °C for 10 min at 300 rpm in mixing block. The MNase digestion reaction was quenched with EDTA at 10 mM final concentration. After 10 min on ice, the nuclei were washed once with 1 ml of RIPA buffer (140 mM NaCl and complete protease inhibitors). Pellets were resuspended in 300 μl of RIPA buffer (140 mM) and sonicated (3 cycles, medium intensity, 30 s on/off intervals) and centrifuged at 18,000 × g, 4 °C for 10 min. DNA was recovered after reverse crosslinking and phenol chloroform extraction. After precipitating and pelleting, DNA was dissolved in 30 μl of TE and resolved on agarose gel. The ~147 bp fragments corresponding to the mono nucleosomal fragments were gel extracted and after library preparation, were subjected to 50 bp paired end sequencing on Illumina HiSeq 2500 platform.

**DamID-Seq.** pUAST- LT3- ORF1 vector (kind gift from A. Brand) was used to clone Cdk9 as a C-terminal Dam-fusion protein. The Dam-Cdk9 itself was cloned downstream of mcherry (as a primary ORF) separated by stop codon. This ensured low level expression of the Dam-Cdk9 fusion protein. S2R+ cells were plated in 10 cm dish and subjected to control and Mago knockdowns using dsRNA, as described earlier. On the sixth day of knockdown pUAST-LT3-Dam-Cdk9 was co-transfected with pActin-Gal4 vector to induce Dam-Cdk9 expression, using effectene transfection reagent according to manufacturer's protocol. The Dam alone control was similarly transfected in control and Mago depleted S2R+ cells. DNA was isolated from cells after 16 h of transfection and subsequent treatments were performed as described[86]. Purified and processed genomic DNA of two biological duplicates was subjected to library preparation using the NebNext DNA Ultra II library kit (New England Biolabs) and sequenced on a NextSeq500.

**Computational analysis.** RNA-Seq: The libraries were sequenced with a read length of 71 bp in paired end mode. Mapping was performed using STAR[87] (v. 2.5.1b) against ENSEMBL release 84. Counts per gene were derived using htseq count (v.0.6.1p1). Differential expression analysis was done using DESeq2[88] (v.1.10.1), differential expressed genes were filtered for an FDR of 1% and a fold change of 1.3. Splicing analysis was done using DEXSeq[89] (v. 1.16.10), and rMATS[90] (v. 3.2.1b) with 10% FDR filtering. Genes were defined as expressed if they had coverage above 1 rpkm in the averaged control samples.

ChIP-Seq: The libraries were sequenced on a HiSeq2500/NextSeq500 in either paired end or single end mode. De-multiplexing and fastq file conversion was performed using blc2fastq (v.1.8.4/v.2.19.1) for all libraries save the Ser2P ChIPs. Ser2P ChIP libraries were de-multiplexed using 6 bp front tags. After sorting, the tags and the A-overhang base were trimmed (7 bp in total). Reads from ChIP-Seq libraries were mapped using bowtie2[91] (v. 2.2.8), and filtered for uniquely mapped reads. The genome build and annotation used for all Drosophila samples was BDGP6 (ensemble release 84). The genome build and annotation used for the HeLa samples was hg38 (ENSEMBL release 84). Peak calling was performed using MACS2[92] (v 2.1.1–20160309). Further processing was done using R and Bioconductor packages. Input normalized bigwig tracks were produced using Deeptools[93]. The spike-in normalization was done according to the Orlando et al.[85]. All the libraries (including input) where calibrated using a normalization

factor; defined as the number of reads mapped to the yeast genome (used as a reference/non-test genome) per million of reads mapped to the *Drosophila* genome. Once the libraries were calibrated according to the respective normalization factor, the enrichment was computed for every condition against their respective input.

To assign the target genes bound by pre-EJC components, peaks were called using MACS2 with 2.0-fold enrichment as cut-off. The resulting peaks were annotated with the ChIPseeker package on Bioconductor, using nearest gene to the peak summit as assignment criteria. The intersection of genes bound by all pre-EJC components, i.e., Mago-HA, Y14-HA, and eIF4A3-HA, was defined as pre-EJC bound.

4sU-Seq: The libraries were sequenced with a read length of 50 bp in single end mode. Mapping was performed using STAR (v. 2.5.1b) and ENSEMBL release 84 for *Drosophila*. Multimapped reads were filtered out, and uniquely mapped reads to the transcript were considered.

Calculation of polymerase release ratio (PRR): PRRs were calculated as follows: for each gene, the TSS region was defined as 250 bp upstream to 250 bp downstream of the TSS. The gene body was defined as 500 bp downstream of the TSS to 500 bp upstream of the TES. The PRR ratio was calculated as the log2 ratio between the enrichment in the downstream region towards the enrichment at the TSS. For each gene, the TSS with the highest average signal around the TSS in the Control condition was selected. Enrichment calculations were based on the enrichment over the input (ChIP-Seq) or t0 (4sU-Seq). The TSS reference was taken from ENSEMBL release 84. Genes with a length smaller than the required length for the calculation were excluded. All libraries containing spike-in controls were normalized to spike-ins following the method described before.

Calculation of elongation rate: For elongation rate calculation, all the genes longer than 10 kb were divided into 100 bp bins (to a total of 20 kb) and the transcriptional wave front was identified in the bin with lowest local minimum signal. The distance in base pairs covered by the wave front between 2 min after DRB removal and 8 min is then divided by the corresponding time interval of 6 min to calculate elongation rates (bp/min).

MNase-Seq: The libraries were sequenced with a read length of 50 bp in paired end mode. De-multiplexing and fastq file conversion was performed using blc2fastq (v.1.8.4). Libraries were de-multiplexed using 6 bp front tags. After sorting, the tags and the A-overhang base were trimmed (7 bp in total). Reads from MNase-Seq libraries were mapped using bowtie2 (v. 2.2.8), and filtered for uniquely mapped reads. The genome build and annotation used for all *Drosophila* samples was BDGP6 (ensemble release 84). Further processing was done using R and Bioconductor packages. Heatmaps and input normalized tracks were produced using Deeptools (v. 2.2.3). Metagene profiles were produced using NGS.plot (v. 2.61).

Targeted DamID-Seq: The libraries were sequenced with a read length of 50 bp in paired end mode. The first read was mapped to *Drosophila melanogaster* genome (BDGP6) using bowtie (v.2.2.9), binned to GATC fragments, and normalized against the Dam-only control[94] using the available damidseq_pipeline on GitHub. The resulting bedgraph files were averaged and smoothened using BEDOPS[95] (v. 2.4.30). The smoothened bedgraph files were converted to bigwig file using SeqPlot, and processed through Deeptools (v. 2.2.3) to generate heatmaps. To quantify the changes at the promoter, the signals in the bedgraph were mapped to the promoters using bedmap tool available in BEDOPS software. The further quantification and plots were generated using R (v 3.4.2), and ggplot2 package available on Bioconductor.

**NELF microarray analysis**. To compare our data with NELF knockdown we used data from this study[60]. The NELF data was retrieved from GEO (GSE20471) and processed in R. Preprocessing was done as described in the original paper. The TSS used for pausing index calculations were determined as the ones with the highest average enrichment $+ -250$ bp around the TSS in the control condition. Calculation of pausing index was done as described in the study.

NELF bound genes were determined as the genes which have an average enrichment >1.3 in the $+ -250$ bp around the TSS (based on the TSS used for pausing index calculations). Any gene ids not matching our reference were converted using the Flybase ID converter tool[96].

**Quantification and statistical analysis**. Statistical parameters and significance are reported in the Figures and the Figure legends. For comparisons of the distribution of different classes we used ANOVA. *t*-Test, two-sample Kolmogorov-Smirnov test and Fisher's test were used for testing the statistical significance. Number of genes used in the box plots is indicated in Supplementary Table 2.

**Reporting Summary**. Further information on experimental design is available in the Nature Research Reporting Summary linked to this Article.

## Data availability

Datasets from RNA-Seq, ChIP-Seq, 4sU-Seq, MNase-Seq, and DamID-Seq have been deposited in NCBI's Gene Expression Omnibus and are accessible through GEO series accession number GSE92389. A Reporting Summary for this Article is available as a Supplementary Information file. All other data supporting the findings of this study are available from the corresponding author upon request.

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

## Acknowledgements

We thank the Bloomington *Drosophila* Stock Center, the Transgenic RNAi Project in Harvard and the Vienna *Drosophila* Resource Center for fly reagents. We also thank Akira Nakamura for the Cdk9 antibody. Support by the IMB Genomics Core Facility and the use of its NextSeq500 (INST 247/870–1 FUGG) is gratefully acknowledged. We also thank the IMB Bioinformatics Core Facility for tremendous support; members of Ulrich lab, especially Lilliana Batista for help with yeast chromatin for "spike-in" control; members of the Roignant lab for fruitful discussion; and Enrico Cannavo, Yad Ghavi-Helm, Guillaume Junion, Jessica Treisman for critical reading of the manuscript. This work was supported by the Marie Curie CIG 334288.

## Author contributions

J.A. and J.Y.R. designed the experiments. J.A performed the experiments and analyzed the data, D.B. helped with RIP experiments. N.K., F.M., J.A., and H.B performed the bioinformatics and statistical analyses. G.K.M. carried out the in vivo and cell culture rescue experiments. J.A. and J.Y.R. wrote the paper.

## Additional information

**Competing interests:** The authors declare no competing interests.

