## [Peer Review File · Nature Communications]

Reviewers' comments:

Reviewer #1 (Remarks to the Author):

In the manuscript titled "Promoter-proximal pausing mediated by the exon junction complex regulates splicing", Akhtar et al. demonstrate a novel non-canonical function of EJC components in Pol II pausing at promoter-proximal regions. By using ChIP-seq and 4sUDRB-seq, authors have shown that the EJC core factors, but not peripheral proteins, can pause Pol II but has no significant effect on Pol II elongation rate at gene body. Further, artificially tethering EJC core components to 5' of nascent transcripts of a reporter induces Pol II pausing. By protein immunoprecipitation, EJC protein, Mago, is shown to interact with pausing Pol II (Ser5P), but not elongating Pol II (Ser2P). Knockdown of EJC components increases the association between Pol II and Cdk9 and Cdk12, two proteins required for transition of Pol II to elongation. Cdk9 and Mago double knockdown partially restores the splicing and photoreceptor defect in Mago knockdown flies. Finally, the authors also show that the role of pre-EJC in promoter proximal pausing is conserved in human cells.

Overall, the manuscript has provided convincing data in support of their major conclusions regarding a novel function of EJC components in pausing Pol II with mechanistic insights. We recommend that the manuscript can be "accepted with minor revision" for publication in Nature Communications. Specific comments are below.

Major comments:

1. Based on similar effects of individual knockdown of Mago, Y14 and eIF4AIII on promoter-proximal pausing the authors draw a major conclusion that this function is performed by the "pre-EJC" assembled from the three factors. However, there is no direct evidence presented that the three proteins indeed function in a complex. The authors can address this by comparing the ability of an EJC-interaction mutant of Mago with the wild-type protein in rescue of splicing and promoter-proximal pausing defects (using MAPK as in Figs 1a and S1b).
2. It will be helpful to comment on the "strength" of Pol II pausing caused by the pre-EJC as compared to more canonical negative regulators such as NELF and DSIF.
3. Fig 1c, d: metagene plots, it is not clear if confidence intervals for the two plots being compared is shown or not. Same applies to some (e.g. Fig 3a, 5a) but not all (e.g., Fig 1e, 1j, 1k) metagene plots in the manuscript.
4. It is difficult to compare heatmaps in Fig 2e and 5c to assess differences that the authors are trying to show. It will be better if fold-change in knockdown experiments as compared to controls are also shown as a heatmap to more clearly show the differences.

Minor comments:

1. Fig S1a: x-axis should include label such as "dsRNA transfected".
2. Fig S1b, c: splicing patterns seen by PCR seem to be different between b and c for the Mago knockdown condition.
3. Fig S1d: From the genome browser screenshot, it is difficult to tell which exons are skipped.
4. Fig S1f and Fig S5d: UpSet plots are difficult to interpret. Consider using Venn diagrams instead.
5. Pg10, lines 11-13. "Consistently, the proportion of Mago or pre-EJC-bound genes was higher at highly expressed genes (Supplementary Fig. 2f and 5i).": no information from Fig S2f is used.
6. In Fig S5j. No p-value is shown in the figure despite the legend.
7. Fig S6a, protein labels are switched in western blot. Same in Fig S6g.
8. Fig7i, the quantitation labels under the image are not aligned with gel lanes.

Reviewer #2 (Remarks to the Author):

Exon Junction Complex (EJC) is deposited on RNA co-transcriptionally and functions to control the

fate of the transcript. EJC's roles beyond this classical model have been proposed in the past. The current manuscript by Akhtar et al add a mechanistic view to the non-classical functions of EJC by showing that (i) EJC regulates promoter-proximal RNA Pol II pausing by restricting Cdk9 activity, and (ii) the pausing-associated function is sufficient to explain the role of EJC in splicing regulation.

While the two claims made by the authors are intriguing and interesting, more rigorous experimental tests need to be performed in order to substantiate the proposed model. With global functions of EJC already known, mechanistic insights must be brought in before arriving at the claims authors make. The three critical issues that would require the authors' attention are described below:

(1) If EJC association to nascent transcript and Pol II was causally associated with restricting Cdk9 activity, then EJC will not be expected to bind highly expressed genes. However, authors themselves point out that EJC binds irrespective of the expression status. Also, it is not clear how EJC 'selects' to bind at certain genes. Both these questions require a deeper understanding of how EJC relates to pausing: as a cause or as a consequence. Experiments with specific inhibitors of initiation, elongation of Pol II, rather than general Amanitin used in the study, may be able to dissect this issue. Using a dynamic expression model of a gene, such as heat-shock genes or ecdysone-responsive loci, will help tease apart the role of EJC. Previously published studies have already performed CHIPseq of EJC along with polytene staining to show that EJC binds to Hsp loci (PMID: 27879206). What happens to Pol II before/ during/ after heat-shock at these loci in absence of EJC will provide an insight. Finally, the experiments describing tethering of EJC to nascent RNA of luciferase is important as it takes apart the myriad other roles of EJC. It will be very useful if this experiment is repeated using another gene model, preferably at an endogenous gene that does not show EJC binding, and does not show pausing (should be identifiable using the genomic datasets). The knock-in of EJC binding site will be immensely useful in arguing against the hypothesis that EJC knock-down causes a generic activation of Cdk9, not just in a locus-specific manner.

(2) EJC's proposed role in pausing is not developed in the context of pre-existing regulators/ known functions. E.g. could EJC recruit NELF, and thus inhibit Cdk9/ enhance pausing? Alternatively, is EJC itself recruited by NELF and both the proteins cooperate to establish pausing? This issue is highly relevant as NELF has also been proposed to bind nascent RNA, possibly competing or cooperating with EJC. NELF mutants/ knock-down, NELF CHIP with/ without EJC function as well as with Cdk9 inhibitors should be able to provide the details linking EJC and pausing. Moreover EJC is known to have a role in mRNA capping. How much of this role is relevant in pausing? An ideal experiment using mutants that are deficient in capping but not in pausing will be useful. But if such mutants do not exist, it will be understandably difficult to delineate the two processes. Nonetheless, mRNA capping and pausing have been linked before, and in the context of the information about EJC/ capping, it will be important to tease them apart.

(3) The authors suggest that the pausing function explains the splicing regulation by EJC. For this claim to be convincing, the authors should use the ectopic recruitment model, with intron-containing reporter gene/ possibly using an endogenous gene with introns (also to answer the issue #2 raised above). The evidence authors use to link pausing and splicing can be explained by various other 'global' functions of EJC, and hence local tethering of EJC will be a useful experiment.

Other issues that must also be taken care of:

(i) EJC binds promoters in an RNase-sensitive manner has been previously reported (PMID: 27879206), and the authors should cite these results appropriately in the intro and results. Obviously the CHIP-binding observed in this manuscript is not at all unexpected, as authors claim (Page 9). On this note, it will be crucial to compare and contrast published dataset of Y14 with the binding profile reported in this manuscript. The published data clearly indicated that eIF4AIII shows a binding profile quite distinct from Mago/ Y14, based on Polytene. Also earlier results show that EJC does not pull down Pol II, unlike what is reported in this manuscript. This discordance may stem from biology, and needs to be discussed/ tested.

(ii) In general the graphs in the manuscript are poorly labeled. All axes must be clearly indicated

(see 1b). Whenever there is 'fold enrichment/ change', it must be made clear what is enriched over what in the figure itself (see 1c for example). For showing ChIP-qPCR data, please refrain from using fold enrichment, rather use '% of input' or some standard form of quantitative ChIP output.

(iii) For metaplots, it is not clear how replicates are dealt with (e.g. 1c/d/e..). Also on page 7, the authors indicate 'equal sized quartiles'...It is not clear on what basis the quartiles were made - based on changes upon knock-down or control values?

(iv) As a key method of genomic analyses, authors have resorted to comparing EJC-bound and unbound genes. It must be made sure that the genes being compared have an overall similar Pol II profile/ amount before knocking down EJC. Else the comparison may reflect an effect of a global change in transcription that depended on starting Pol II levels. Such analyses will then be a mis-interpretation of the data. In this regard, it will be easier if the authors showed real distributions in two conditions, rather than showing ratios/ fold enrichment, which hides the differences in basal levels.

(v) Page 9: The degree of overlap between...high (34%). This is not very high if you consider that you ChIP'ed three proteins in the same complex! See comment (i) regarding previously reported overlap.

(vi) Wrong labeling in S6a.

Reviewer #3 (Remarks to the Author):

The Exon Junction Complex (EJC) has recently been shown to facilitate splicing in flies, but the mechanism by which the EJC helps splicing has been elusive.

Akhtar et al., reports that the EJC ensures correct splicing by preventing the premature release of RNA polymerase II (PolII) from the promoter.

The following are the major claims of the paper:

1. The depletion of the core EJC factors leads to a decreased PolII occupancy at gene promoters, which is a consequence of reduced pausing rather than a decrease in transcription initiation.
2. The core EJC factors, but not the accessory co-factor RnpS1, occupy promoter regions independently of splicing, but dependently of the interaction with PolII mediated by nascent transcripts.
3. The EJC restricts the recruitment of P-TEFb to the promoter region.
4. A co-depletion of CDK9 or CDK12 rescues the MAPK exon exclusion phenotype and the eye development defect caused by EJC depletion.

It is well accepted that splicing occurs co-transcriptionally. But, the mechanism by which PolII influences the splicing outcomes remains unclear. It is known that the EJC physically associates with other splicing factors including SR proteins. Therefore, it was postulated that the EJC facilitates splicing by recruiting splicing factors on pre-mRNA. This work demonstrated that the EJC aids splicing at least partially by preventing the release of PolII from the promoter, thus shedding light on a previously undescribed function of the EJC. Although the manuscript does not fully answer the question of why long-intron-containing genes are more sensitive to the loss of EJC, it advances our understanding of how the assembly of PolII elongation complex at the promoter influences co-transcriptional RNA processing.

Although some parts need clarification, their claims are overall supported by the data. Therefore, I recommend a publication in Nature Communications.

The following are the suggestions for improvement:

Major comments:

1. For all the box plots comparing pre-EJC bound and unbound genes, how were the unbound

genes selected? Was the expression cut-off comparable to the pre-EJC bound genes? Otherwise, that would affect the comparison regardless of the binding status of pre-EJC.

2. The rescue experiments are remarkable. If the depletion of Cdk12 rescues the EJC-KD phenotype, do we not expect to see the same when Paf1 complex components are additionally depleted because Paf1 complex has been shown to recruit Cdk12 to the paused polII? Can authors also show by RT-PCR that the depletion of Paf1 complex in addition to the EJC-KD does not impact the MAPK splicing because RT-PCR appears to be a more quantitative measurement than eye phenotype?

3. To further strengthen the Author's findings, does the depletion of NELF lead to MAPK mis-splicing phenotype (and the eye developmental defect)?

Minor comments:

1. Figure labelling needs improvements. They should be understandable without having to read legend or method section.

2. Where were Transcription Start Sites taken from?

3. The number of genes in each category should be noted for box plots.

4. Some comparisons lack an assessment of statistical significance, e.g. Fig4a-c.

5. In ChIP-seq tracks such as Fig1b, Y-axis lacks labeling.

6. How is fold change in ChIP-seq experiments calculated in Figs 1 and 2? Is it normalised to the total genome mappers divided by the cloning counts of spikes?

7. How were the BoxB-lambdaN constructs made? Were the expression levels comparable? Was GFP fused to the nuclear localisation signal in order for it to be comparable to splicing factors?

8. Fig1a: Fold enrichment over what?

9. Fig1b: Isn't it strange to see a complete depletion of PolII signal from the gene body in the control KD?

10. Fig1g: What does the x-axis mean?

11. Why is FigS2a so different from Fig1a?

12. Fig4h: Why not showing the same Venn diagram as e-g?

13. Fig6e: The standard deviation may not be calculated from two data points.

First of all, we would like to thank the reviewers for the constructive comments. Our replies are given below.

Reviewer #1 (Remarks to the Author):

In the manuscript titled "Promoter-proximal pausing mediated by the exon junction complex regulates splicing", Akhtar et al. demonstrate a novel non-canonical function of EJC components in Pol II pausing at promoter-proximal regions. By using ChIP-seq and 4sUDRB-seq, authors have shown that the EJC core factors, but not peripheral proteins, can pause Pol II but has no significant effect on Pol II elongation rate at gene body. Further, artificially tethering EJC core components to 5' of nascent transcripts of a reporter induces Pol II pausing. By protein immunoprecipitation, EJC protein, Mago, is shown to interact with pausing Pol II (Ser5P), but not elongating Pol II (Ser2P). Knockdown of EJC components increases the association between Pol II and Cdk9 and Cdk12, two proteins required for transition of Pol II to elongation. Cdk9 and Mago double knockdown partially restores the splicing and photoreceptor defect in Mago knockdown flies. Finally, the authors also show that the role of pre-EJC in promoter proximal pausing is conserved in human cells. Overall, the manuscript has provided convincing data in support of their major conclusions regarding a novel function of EJC components in pausing Pol II with mechanistic insights. We recommend that the manuscript can be "accepted with minor revision" for publication in Nature Communications. Specific comments are below.

Major comments:

1. Based on similar effects of individual knockdown of Mago, Y14 and eIF4AIII on promoter-proximal pausing the authors draw a major conclusion that this function is performed by the "pre-EJC" assembled from the three factors. However, there is no direct evidence presented that the three proteins indeed function in a complex. The authors can address this by comparing the ability of an EJC-interaction mutant of Mago with the wild-type protein in rescue of splicing and promoter-proximal pausing defects (using MAPK as in Figs 1a and S1b).

As suggested by the reviewer we cloned a mutant for Mago that has lost its ability to interact with other EJC components. The point mutation was based on the crystal structure previously published in (Fribourg et al, 2003), which was shown to alter the interaction of Mago with Y14. While the WT Mago was able to rescue *MAPK* splicing (Fig. S1d) and the promoter-proximal pausing (Fig. S2a, b), these defects were not rescued with the Mago mutant construct. This was not due to destabilization of the mutant protein, as its level was comparable to the WT construct (Fig. S1e). Therefore pre-EJC assembly is required both for *MAPK* splicing and the regulation of promoter proximal pausing. These new data have now been included in the revised version.

Fribourg S, Gatfield D, Izaurralde E, Conti E (2003) A novel mode of RBD-protein recognition in the Y14-Mago complex. Nature structural biology 10: 433-439

2. It will be helpful to comment on the "strength" of Pol II pausing caused by the pre-EJC as compared to more canonical negative regulators such as NELF and DSIF.

We have now analyzed the publicly available NELF dataset (Gilchrist et al, 2010), and compared the effect of the NELF knockdown with the pre-EJC KD. We have added a new supplementary figure (Fig. S8) to illustrate this analysis.

- First we noticed that NELF binds substantially more genes in comparison to the pre-EJC (3796 versus 816). 55 % of pre-EJC-bound genes overlap with NELF binding, 45 % do not (Fig. S8a).
- Their binding to promoters seems largely independent of each other (Fig. S8h-j)
- As shown for pre-EJC components, NELF binding was also overrepresented in the quartile containing genes with the strongest expression (Fig. S8b, c) and highest pause index (Fig. S8d-e).
- Accordingly, the fold change in Pol II pausing at the TSS upon NELF KD was higher on highly paused genes and comparable in its strength to the pre-EJC KD (Fig. S8f, g).
- NELF does not bind *MAPK* and is not required for its splicing (Fig. S8k).

These results indicate that the effect of NELF and pre-EJC components on Pol II pausing bears some similarities. Nevertheless the binding of the pre-EJC to promoters is much more restricted and seems independent on the presence of NELF. Conversely NELF does not require the pre-EJC to bind chromatin. These results have now been included in a new supplementary figure S8.

Gilchrist DA, Dos Santos G, Fargo DC, Xie B, Gao Y, Li L, Adelman K (2010) Pausing of RNA polymerase II disrupts DNA-specified nucleosome organization to enable precise gene regulation. Cell 143: 540-551

3. Fig 1c, d: metagene plots, it is not clear if confidence intervals for the two plots being compared is shown or not. Same applies to some (e.g. Fig 3a, 5a) but not all (e.g., Fig 1e, 1j, 1k) metagene plots in the manuscript.

All the deeptools plots have now been modified to display the standard error of the mean, consistent with other metagene plots.

4. It is difficult to compare heatmaps in Fig 2e and 5c to assess differences that the authors are trying to show. It will be better if fold-change in knockdown experiments as compared to controls are also shown as a heatmap to more clearly show the differences.

We now provide heatmaps for Fig 2e and 5c to better highlight the differences. To validate the increased occupancy of Cdk9 in the absence of Mago we also performed ChIP-qPCR and monitored the binding of Cdk9-HA at the *MAPK* locus. According to the DamID result, we found that Cdk9 occupancy was specifically increased at the 5' end of the *MAPK* gene (Fig. 5d).

Minor comments:

1. Fig S1a: x-axis should include label such as "dsRNA transfected".

The figure has been modified in the revised version.

2. Fig S1b, c: splicing patterns seen by PCR seem to be different between b and c for the Mago knockdown condition.

The extent of *MAPK* splicing defect is sensitive to the level of knockdown and the passage number of the cells. Furthermore the gel running time can also slightly alter the visualization of the splicing pattern. Although we tried to perform the experiments in identical conditions we still observed some variability. However we do not think that this affects the interpretation of the results.

3. Fig S1d: From the genome browser screenshot, it is difficult to tell which exons are skipped.

The figure has been modified to include a colored rectangle highlighting the skipped exons.

4. Fig S1f and Fig S5d: UpSet plots are difficult to interpret. Consider using Venn diagrams instead.

The revised figures now represent the data with venn diagram.

5. Pg10, lines 11-13. "Consistently, the proportion of Mago or pre-EJC-bound genes was higher at highly expressed genes (Supplementary Fig. 2f and 5i)." no information from Fig S2f is used.

We have corrected this in the revised version.

6. In Fig S5j. No p-value is shown in the figure despite the legend.

We have included the statistics in the revised figure.

7. Fig S6a, protein labels are switched in western blot. Same in Fig S6g.

We thank the reviewer for pointing this out. The labels have been corrected in the revised version.

8. Fig7i, the quantitation labels under the image are not aligned with gel lanes.

We have corrected the quantitation labels to align with the gel lanes.

Reviewer #2 (Remarks to the Author):

Exon Junction Complex (EJC) is deposited on RNA co-transcriptionally and functions to control the fate of the transcript. EJC's roles beyond this classical model have been proposed in the past. The current manuscript by Akhtar et al add a mechanistic view to the non-classical functions of EJC by showing that (i) EJC regulates promoter-proximal RNA Pol II pausing by restricting Cdk9 activity, and (ii) the pausing-associated function is sufficient to explain the role of EJC in splicing regulation.

While the two claims made by the authors are intriguing and interesting, more rigorous experimental tests need to be performed in order to substantiate the proposed model. With global functions of EJC already known, mechanistic insights must be brought in before arriving at the claims authors make. The three critical issues that would require

the authors' attention are described below:

If EJC association to nascent transcript and Pol II was causally associated with restricting Cdk9 activity, then EJC will not be expected to bind highly expressed genes. However, authors themselves point out that EJC binds irrespective of the expression status. Also, it is not clear how EJC 'selects' to bind at certain genes. Both these questions require a deeper understanding of how EJC relates to pausing: as a cause or as a consequence.

We realized we did not explain very well this part. Although we propose that there is a potential competition between pre-EJC components and Cdk9 binding, we show that nascent transcription is essential for pre-EJC components binding and therefore one would expect that the pre-EJC binds more tightly highly expressed genes in comparison to genes with low expression. Indeed, we found that pre-EJC-bound genes are overrepresented for highly expressed genes (see Fig. 2f and S6k). Currently, the dynamics of pre-EJC components and Cdk9 binding is unknown and their understanding will require additional work.

Experiments with specific inhibitors of initiation, elongation of Pol II, rather than general Amanitin used in the study, may be able to dissect this issue. Using a dynamic expression model of a gene, such as heat-shock genes or ecdysone-responsive loci, will help tease apart the role of EJC. Previously published studies have already performed ChIPseq of EJC along with polytene staining to show that EJC binds to Hsp loci (PMID: 27879206). What happens to Pol II before/ during/ after heat-shock at these loci in absence of EJC will provide an insight.

As suggested, we have performed ChIP experiments to examine binding of Mago-HA after treating the cells with specific inhibitors of Pol II initiation (triptolide) or elongation (DRB). We first confirmed the efficiency of the drug treatment on Pol II binding (Fig. S6i). The treatment with the initiation specific inhibitor triptolide lead to loss of Mago binding at the target genes while the elongation specific inhibitor DRB had no significant effect on Mago binding (Fig. S6j). These results confirm that nascent transcription is required for Mago binding. These experiments are now included in the revised version.

We also performed ChIP for Pol II on the *Hsp70* gene before, during and after heatshock (HS) treatment of S2R+ cells. The "before HS" sample was not subjected to HS, the samples labeled "during HS" was subjected to a 30-minutes HS at 37° C and fixed during the HS, while the samples "after HS" was allowed to recover for 30 minutes before fixation. We found that Mago KD results in lower promoter Pol II occupancy in the steady state condition (before the HS), as shown by the qPCR result (Fig. S4). During the HS there was no significant differences in Pol II occupancy in Mago KD compared to the control. However, after HS, Pol II pausing was not maintained comparable to the control. These results suggest a critical function for the EJC in maintaining Pol II pausing during the steady state condition and during HS recovery. These new data are discussed in the manuscript.

Finally, the experiments describing tethering of EJC to nascent RNA of luciferase is important as it takes apart the myriad other roles of EJC. It will be very useful if this experiment is repeated using another gene model, preferably at an endogenous gene that does not show EJC binding, and does not show pausing (should be identifiable using

the genomic datasets). The knock-in of EJC binding site will be immensely useful in arguing against the hypothesis that EJC knock-down causes a generic activation of Cdk9, not just in a locus-specific manner.

We have repeated the EJC-tethering experiment using additional genes. The genes that were included for this experiment were *piwi*, *Crk*, and *BBS8*. *piwi* is normally not expressed in S2R+ cells but is well studied for Mago-dependent splicing defect in a minigene context (see (Hayashi et al, 2014; Malone et al, 2014)). The *Crk* and *Bbs8* are not bound by EJC but *BBS8* display Pol II pausing while *Crk* does not. By ectopically tethering Mago to the 5' end of these mRNA we observed increase in pausing for all these genes, as shown by Pol II ChIP-qPCR experiment (Fig. S7a). These results indicate that tethering Mago at the 5' end of an RNA, irrespective of whether it is normally bound by the EJC, is sufficient to increase Pol II pausing.

Hayashi R, Handler D, Ish-Horowicz D, Brennecke J (2014) The exon junction complex is required for definition and excision of neighboring introns in Drosophila. Genes & development 28: 1772-1785

Malone CD, Mestdagh C, Akhtar J, Kreim N, Deinhard P, Sachidanandam R, Treisman J, Roignant JY (2014) The exon junction complex controls transposable element activity by ensuring faithful splicing of the piwi transcript. Genes & development 28: 1786-1799

(2) EJC's proposed role in pausing is not developed in the context of pre-existing regulators/ known functions. E.g. could EJC recruit NELF, and thus inhibit Cdk9/ enhance pausing? Alternatively, is EJC itself recruited by NELF and both the proteins cooperate to establish pausing? This issue is highly relevant as NELF has also been proposed to bind nascent RNA, possibly competing or cooperating with EJC. NELF mutants/ knock-down, NELF ChIP with/ without EJC function as well as with Cdk9 inhibitors should be able to provide the details linking EJC and pausing. Moreover EJC is known to have a role in mRNA capping. How much of this role is relevant in pausing? An ideal experiment using mutants that are deficient in capping but not in pausing will be useful. But if such mutants do not exist, it will be understandably difficult to delineate the two processes. Nonetheless, mRNA capping and pausing have been linked before, and in the context of the information about EJC/ capping, it will be important to tease them apart.

We have performed a series of experiments and computational work to investigate a possible interplay between NELF and the pre-EJC. See also response to reviewer 1. The results are now included in a new supplementary figure (Fig. S8) and are summarized below:

- We found that NELF binds substantially more genes than the pre-EJC (3796 versus 816) and their binding only partially overlaps.
- Both NELF and pre-EJC binding are enriched on highly expressed genes.
- NELF and pre-EJC binding are enriched on genes that are strongly paused in WT condition. Furthermore, the changes on Pol II pausing after their depletion is higher on this class of strongly paused genes.
- Depletion of Mago has only minor effect on NELF binding. Similarly, depletion of NELF only slightly affects Mago binding. These results suggest that their binding is

- largely independent of each other. In comparison, depletion of Cap binding protein (Cbp), which affects RNA stability, has more pronounced effect on pre-EJC binding.
- NELF does not bind *MAPK* and its KD has no effect on *MAPK* splicing (Fig S8i).

In conclusion our experiments suggest that EJC binding is restricted to specific genes unlike NELF, and its binding is not dependent on NELF recruitment. More experiments will be required to better understand how their binding to promoters is regulated.

(3) The authors suggest that the pausing function explains the splicing regulation by EJC. For this claim to be convincing, the authors should use the ectopic recruitment model, with intron-containing reporter gene/ possibly using an endogenous gene with introns (also to answer the issue #2 raised above). The evidence authors use to link pausing and splicing can be explained by various other 'global' functions of EJC, and hence local tethering of EJC will be a useful experiment.

We have repeated the tethering experiment as explained above. We examined splicing of these minigene constructs, upon Mago KD and in condition where ectopic Mago was tethered to the 5' end of these genes. We demonstrate that tethering Mago could rescue the Mago dependent splicing defect for both *piwi* and *Crk* minigene constructs (Fig. S7b), arguing that the increase in promoter proximal pausing induced by tethering Mago at 5' end of the genes is sufficient to rescue the splicing defects.

Other issues that must also be taken care of:

(i) EJC binds promoters in an RNase-sensitive manner has been previously reported (PMID: 27879206), and the authors should cite these results appropriately in the intro and results.

This finding is now properly cited in the introduction and in the result section.

Obviously the ChIP-binding observed in this manuscript is not at all unexpected, as authors claim (Page 9). On this note, it will be crucial to compare and contrast published dataset of Y14 with the binding profile reported in this manuscript. The published data clearly indicated that eIF4AIII shows a binding profile quite distinct from Mago/ Y14, based on Polytene. Also earlier results show that EJC does not pull down Pol II, unlike what is reported in this manuscript. This discordance may stem from biology, and needs to be discussed/ tested.

We have now compared the published Y14 binding profile with the profile of pre-EJC components that we produced. While there is an extensive overlap (Fig. 1a, below), the Y14 ChIP identified many more targets. We have two explanations for this discrepancy. On one hand we are very conservative as we considerer positive targets only when the three pre-EJC components bind at the same position. On the other hand we have previously produced a ChIP profile using our endogenous anti-Mago antibody. In this case we identified many more targets (Fig. 1b, below). However, we believe that many of these targets were false positives as the signal remained unchanged upon Mago KD, even despite the fact that this antibody gives specific signal on Western blot. It has been previously reported that many antibodies used in ChIP unspecifically bind at open chromatin regions that strongly correlates with active promoters. These peaks have been called phantom peaks (Jain et al, 2015). Indeed, the peaks from our initial Mago

ChIP strongly correlate with highly expressed genes, which have higher open chromatin at promoters. In conclusion, based on our experience and the reported caveat we believe that each ChIP should be validated by a respective KD of the bait, especially when peaks are mainly localized at promoter regions.

Fig 1. The overlap of EJC ChIP with publicly available Y14 ChIP dataset **(a)** Venn diagram showing the extent of overlap of EJC ChIP with Y14 Public dataset. **(b)** Venn diagram showing the extent of overlap of Mago ChIP performed with rabbit antibody directed against endogenous Mago (not included in the manuscript) with Y14 Public dataset.

Regarding the interaction of Mago with Pol II, we initially missed it because we used an antibody against Ser2P. We believe that the interaction is highly specific to Ser5P.

Jain, D., Baldi, S., Zabel, A., Straub, T. & Becker, P.B. Active promoters give rise to false positive 'Phantom Peaks' in ChIP-seq experiments. Nucleic Acids Res 43, 6959-68 (2015).

(ii) In general the graphs in the manuscript are poorly labeled. All axes must be clearly indicated (see 1b). Whenever there is 'fold enrichment/ change', it must be made clear what is enriched over what in the figure itself (see 1c for example). For showing ChIP-qPCR data, please refrain from using fold enrichment, rather use '% of input' or some standard form of quantitative ChIP output.

We have better annotated our ChIP-qPCR enrichments in the revised figures. We report ChIP-qPCR enrichment as fold enrichment as it is two-fold normalization against a negative loci as well as input. In our experience this representation is far more robust and mitigates the effect of slight variation in starting amount and recovery (due to beads or antibody lots). We always start our ChIP-qPCR experiments with equal amount of chromatin that is determined by fluorometric quantification.

(iii) For metaplots, it is not clear how replicates are dealt with (e.g. 1c/d/e..). Also on page 7, the authors indicate 'equal sized quartiles'...It is not clear on what basis the quartiles were made - based on changes upon knock-down or control values?

Metagene profile plots are created from input and spike in normalised coverage files which are averaged for the two replicates. The standard error of the mean shading shows the standard error of the mean based on the genes shown.

PRR Quartiles were made based on the replicate averaged PRR values of Ctrl_Pol II pulldown, for which the TSSs are bound in both Ctrl_Pol II and Mago_Pol II. The data was divided into Quartiles and assigned accordingly.

(iv) As a key method of genomic analyses, authors have resorted to comparing EJC-bound and unbound genes. It must be made sure that the genes being compared have an overall similar Pol II profile/ amount before knocking down EJC. Else the comparison may reflect an effect of a global change in transcription that depended on starting Pol II levels. Such analyses will then be a mis-interpretation of the data. In this regard, it will be easier if the authors showed real distributions in two conditions, rather than showing ratios/ fold enrichment, which hides the differences in basal levels.

Although Mago binding shows a slight preferential enrichment for the lowest PRR quartile (highly paused genes), the Pol II signal distribution around the TSS for EJC bound and unbound categories was comparable before Mago depletion, ensuring the validity of our analysis. See also answer to reviewer 3.

(v) Page 9: The degree of overlap between...high (34%). This is not very high if you consider that you ChIP'ed three proteins in the same complex! See comment (i) regarding previously reported overlap.

We agree with the reviewer. We have modified this sentence as “The degree of overlap between the bound targets of pre-EJC components is of 34%, corresponding to 816 genes.”

(vi) Wrong labeling in S6a.

This has been corrected in the revised version.

Reviewer #3 (Remarks to the Author):

The Exon Junction Complex (EJC) has recently been shown to facilitate splicing in flies, but the mechanism by which the EJC helps splicing has been elusive. Akhtar et al., reports that the EJC ensures correct splicing by preventing the premature release of RNA polymerase II (PolII) from the promoter.

The following are the major claims of the paper:

1. The depletion of the core EJC factors leads to a decreased PolII occupancy at gene promoters, which is a consequence of reduced pausing rather than a decrease in transcription initiation.
2. The core EJC factors, but not the accessory co-factor RnpS1, occupy promoter regions independently of splicing, but dependently of the interaction with PolII mediated by nascent transcripts.
3. The EJC restricts the recruitment of P-TEFb to the promoter region.
4. A co-depletion of CDK9 or CDK12 rescues the MAPK exon exclusion phenotype and the eye development defect caused by EJC depletion.

It is well accepted that splicing occurs co-transcriptionally. But, the mechanism by which PolIII influences the splicing outcomes remains unclear. It is known that the EJC physically associates with other splicing factors including SR proteins. Therefore, it was postulated that the EJC facilitates splicing by recruiting splicing factors on pre-mRNA. This work demonstrated that the EJC aids splicing at least partially by preventing the release of PolIII from the promoter, thus shedding light on a previously undescribed function of the EJC. Although the manuscript does not fully answer the question of why long-intron-containing genes are more sensitive to the loss of EJC, it advances our understanding of how the assembly of PolIII elongation complex at the promoter influences co-transcriptional RNA processing.

Although some parts need clarification, their claims are overall supported by the data. Therefore, I recommend a publication in Nature Communications.

The following are the suggestions for improvement:

Major comments:

1. For all the box plots comparing pre-EJC bound and unbound genes, how were the unbound genes selected? Was the expression cut-off comparable to the pre-EJC bound genes? Otherwise, that would affect the comparison regardless of the binding status of pre-EJC.

The numbers of genes included in different categories are now provided in a Table 2.

We have compared the distribution of Pol II signal at the TSS in control condition separated according to Mago binding (attached Fig. 2 below).

Fig 2. The Pol II signal distribution of bound and unbound classes in control condition **(a)** The Pol II signal density plot for Mago bound and unbound classes in control condition. The X-axis depicts normalized read per million (rpm) at the TSS ± 250 bp, and the Y-axis shows distribution of the signal. **(b)** The Pol II signal density plot for NELF bound and unbound classes in control condition (Gilchrist DA et al; 2010). The X-axis depicts normalized read per million (rpm) at the TSS ± 250 bp, and the Y-axis shows distribution of the signal.

The signal distribution in the control condition is comparable between the Mago bound and unbound class, the slight shift in distribution could be explained by the pre-disposal of EJC to bind highly paused genes. In comparison the shift in the distribution of NELF-bound and unbound classes, in the publicly available dataset, is slightly higher.

Fig. 4D, S5ij contain PRR values for the genes which are bound in Ctrl_PolIII, Mago_PolIII (intersected Peak calls of replicates overlap at least one TSS of the gene). And then separated after EJC binding.

2. The rescue experiments are remarkable. If the depletion of Cdk12 rescues the EJC-KD phenotype, do we not expect to see the same when Paf1 complex components are additionally depleted because Paf1 complex has been shown to recruit Cdk12 to the paused polIII? Can authors also show by RT-PCR that the depletion of Paf1 complex in addition to the EJC-KD does not impact the MAPK splicing because RT-PCR appears to be a more quantitative measurement than eye phenotype?

As suggested by the reviewer we checked the effect of Paf1 depletion on *MAPK* splicing. In agreement with the lack of rescue of the eye phenotype we found that depletion of components of Paf1 did not rescue *MAPK* splicing. The interpretation of this experiment is not obvious as it is unknown whether Paf1 plays any role in promoter proximal pausing in *Drosophila*. Even in the case of the Vertebrate Paf1 its effect on promoter proximal pausing appears to be cell type specific, making unclear how Paf1 generally regulates this process.

Fig 3. Paf1 knockdown fails to rescue Mago-dependent *MAPK* splicing defects. Agarose gel picture of semi-quantitative RT-PCR for *MAPK* transcripts using RNA from S2R+ cells in the indicated knockdowns. Exons included in wild type condition and skipped in Mago knockdown condition is highlighted by green and red asterisks, respectively.

3. To further strengthen the Author's findings, does the depletion of NELF lead to *MAPK* mis-splicing phenotype (and the eye developmental defect)?

Please see answer to reviewers 1 and 2. NELF does not bind *MAPK* and its depletion has no effect on *MAPK* splicing (Fig. S8k).

Minor comments:

1. Figure labelling needs improvements. They should be understandable without having to read legend or method section.

We have improved the labeling of all the figures.

2. Where were Transcription Start Sites taken from?

Transcription start sites are taken from the ENSEMBL annotation release 84 of *Drosophila melanogaster*. For PRR calculations genes were filtered to have a sufficient length for the PRR calculations (at least 1kb). The method section was extended to include this information.

3. The number of genes in each category should be noted for box plots.

We have included the number of genes included in all the categories shown in the box plots in table 2.

4. Some comparisons lack an assessment of statistical significance, e.g. Fig4a-c.

We have now performed Anova statistical test for all the plots and included the test of statistical significance in the revised version.

5. In ChIP-seq tracks such as Fig1b, Y-axis lacks labeling.

The revised figure now depicts the labeled Y-axis.

6. How is fold change in ChIP-seq experiments calculated in Figs 1 and 2? Is it normalised to the total genome mappers divided by the cloning counts of spikes?

The libraries were normalized by scaling them to the counts of spikeINs, as described earlier (Orlando et al, 2014). Foldchanges are then calculated towards the respective input.

Orlando, D.A. et al. Quantitative ChIP-Seq normalization reveals global modulation of the epigenome. Cell Rep 9, 1163-70 (2014).

7. How were the BoxB-lambdaN constructs made? Were the expression levels comparable? Was GFP fused to the nuclear localisation signal in order for it to be comparable to splicing factors?

The lambdaN and Box-B constructs are derived from the plasmids described earlier (Hilgers et al, 2012). The lambdaN constructs were made by cloning different CDS in frame with LambdaN at the C-terminal. We could not directly test the expression of these constructs because of the lack of antibody against lambdaN. Nevertheless they are all expressed under the control of the same promoter (*actin*) and we always included a transfection control plasmid (Renilla luciferase) to control for transfection variability. Furthermore, we could verify that the LambdaN-GFP fusion protein localizes in the nucleus (albeit some fluorescence was also observed in the cytoplasm). Lastly the opposite effect on Pol II pausing mediated by the tethering of RnpS1 strongly argues that not all RNA binding proteins are behaving similarly with respect to modulation of Pol II pausing.

Hilgers, V., Lemke, S.B. & Levine, M. ELAV mediates 3' UTR extension in the Drosophila nervous system. Genes Dev 26, 2259-64 (2012).

8. Fig1a: Fold enrichment over what?

The fold enrichment was calculated over input after normalizing against a negative locus. In the revised figure the Y-axis is now clearly labeled 'Fold Enrichment over Input.'

9. Fig1b: Isn't it strange to see a complete depletion of PolII signal from the gene body in the control KD?

The total Pol II antibodies we have tested so far always give signal predominantly at the promoter. Nevertheless, now we have zoomed into the genome browser itself and show the signal instead of masking the flanking regions in the illustrator.

10. Fig1g: What does the x-axis mean?

The figure 1g is empirical cumulative distribution function (ECDF) plot and the X-axis shown percentage distribution of genes with a particular PRR level. Now the X-axis is clearly labeled in the revised figure.

11. Why is FigS2a so different from Fig1a?

The difference in the level of enrichment is due to the lot of polyclonal antibody against Ser2P. The Fig. 1a shows the enrichment as a line plot that is easier to see for two conditions. The Fig. S2a shows the enrichment as histogram, as the line plot for various conditions are more difficult to read.

12. Fig4h: Why not showing the same Venn diagram as e-g?

Now we have combined the up and down-regulated nascent RNA expression into one category and display the data consistent with other Venn diagrams in the figure.

13. Fig6e: The standard deviation may not be calculated from two data points.

We completely agree that the standard deviation cannot be calculated from two data points. However the Fig. 6e includes two independent biological replicates with three technical replicates each. The standard deviation displayed in the graph is thus derived from 6 data points, showing biological as well as technical variance.

REVIEWERS' COMMENTS:

Reviewer #1 (Remarks to the Author):

In the revised manuscript, the authors have conscientiously addressed majority of comments from reviewers. They have now provided more experimental evidence to support their major conclusions. However, there are several minor inaccuracies and inconsistencies in the manuscript, which should be fixed before publication. We recommend the authors to make changes to address the following minor issues:

1. Line 114: the details of splicing changes in Btz depletion are not present in Suppl. Fig. 1h, unlike as stated by the authors.
2. Lines 144-145: the Pol II occupancy at TES does not increase upon Y14 depletion in comparison to Mago and eIF4AIII depletion (Suppl Fig. 2e). However, the authors incorrectly describe this result to be similar to Mago depletion.
3. Lines 156-157: Here is what we understand regarding how the authors performed the analysis in Fig. 1h. The four quartiles were defined based on PRR in wild-type conditions. Then the authors looked at fold-change in PRR in the four quartiles, and that is what is shown in Fig. 1h. If that is the case, "upon Mago KD" should be removed from the sentence on line 156 as it makes the wording confusing.
4. Lines 192-194: In the conclusion from the experiment examining the hsp70 during heat shock, the authors should clarify what they mean by "maintain pausing" and "initiate pausing". Also, is the increase in Pol II occupancy in HS conditions (in control and Mago KD) simply due to increased transcription? If this is the case, this occupancy is distinct from that observed due to pausing (and hence the differences seen in no HS or after HS as compared to HS). This should be more clearly stated.
5. Lines 221-224: The authors again incorrectly describe their findings from Suppl. Fig. 6g in this sentence. There seems to be significant difference in Mago-HA ChIP signal upon Cwc22 KD or isoginkgetin treatment as compared to control. The differences are particularly prominent in the case of intron-containing genes. Perhaps, there could be some cross-talk between transcription and splicing dependent EJC factor recruitment mechanisms.
6. Line 226: It should be Suppl. Fig. 6h and not 5h.
7. Suppl. Fig 7a: The authors should include a sentence on why they see a reduction in Pol II occupancy on 3' end of piwi, Crk3 and BBS8 genes but such a change is not see in Luciferase construct (Fig. 3h).
8. Suppl. Fig 7b is not referenced in the main text.
9. Suppl Fig. 8b is similar (the same?) as Suppl Fig. S6k. However, the former is labeled to show "pre-EJC bound" whereas the latter shows "Mago bound".
10. Line 346: Should the reference be to Suppl Fig. 9c instead of Suppl Fig. 7c?
11. Line 349: Should the reference be to Suppl Fig. 7b instead of Suppl Fig. 6b?

Reviewer #2 (Remarks to the Author):

The authors have thoroughly revised the manuscript and answered all the questions raised. I recommend this manuscript to be accepted.

Reviewer #3 (Remarks to the Author):

Authors have answered most of the key questions raised by reviewers. However, I believe several points should be made clear for data presentation before publication.

1. It is not conventional to draw error bars from two biological data points. This applies to Figure 6e and most of meta gene plots comparing control KD and mago(h) KDs. Please display the standard deviation as confidence intervals in selected single libraries of each KD condition. Data

deviation between two replicates should be shown in a different format if necessary.

2. Please provide all the original western blot images in a separate supplementary file.
3. It is still not clear how CHIP-seq data are normalised. How can libraries be normalised both by spike-in and input? Please elaborate the method section rather than just mentioning which tools were used.
4. Fig. 2a-c and S6a, c: no genome-wide profile of RnpS1-HA ChIP is shown.
5. Please explain what x and y axes indicate and what individual dots represent in Figure S2c.
6. Box plots displaying the polymerase release ratio are difficult to read. Please use the full term whenever space is allowed and consider using "paused" and "less paused" with a triangle indicator instead of using low to high or quar. 1 to 4.

Reviewer #1 (Remarks to the Author):

In the revised manuscript, the authors have conscientiously addressed majority of comments from reviewers. They have now provided more experimental evidence to support their major conclusions. However, there are several minor inaccuracies and inconsistencies in the manuscript, which should be fixed before publication. We recommend the authors to make changes to address the following minor issues:

1. Line 114: the details of splicing changes in Btz depletion are not present in Suppl. Fig. 1h, unlike as stated by the authors.

This has now been fixed. It should be read "Supplementary Fig. 1h and data not shown"

2. Lines 144-145: the Pol II occupancy at TES does not increase upon Y14 depletion in comparison to Mago and eIF4AIII depletion (Suppl Fig. 2e). However, the authors incorrectly describe this result to be similar to Mago depletion.

It should be now read: "Similar changes in Pol II occupancy were observed upon depletion of Y14 and eIF4AIII, especially at the TSS, but neither of RnPS1 nor or Btz"

3. Lines 156-157: Here is what we understand regarding how the authors performed the analysis in Fig. 1h. The four quartiles were defined based on PRR in wild-type conditions. Then the authors looked at fold-change in PRR in the four quartiles, and that is what is shown in Fig. 1h. If that is the case, "upon Mago KD" should be removed from the sentence on line 156 as it makes the wording confusing.

This has now been removed.

4. Lines 192-194: In the conclusion from the experiment examining the hsp70 during heat shock, the authors should clarify what they mean by "maintain pausing" and "initiate pausing". Also, is the increase in Pol II occupancy in HS conditions (in control and Mago KD) simply due to increased transcription? If this is the case, this occupancy is distinct from that observed due to pausing (and hence the differences seen in no HS or after HS as compared to HS). This should be more clearly stated.

We agree that during HS condition the increase of Pol II occupancy results from increased transcription, and this occurs in both WT and Mago KD conditions. This suggests that Mago is indeed not involved in transcription initiation but only in pausing. Therefore the distinction we made between initiating and maintaining pausing might be misleading.

We modified this paragraph:

"During HS, Pol II occupancy rose dramatically and the extent of this increase was similar in control versus Mago KD. However, during recovery after HS, Pol II occupancy remained high at the 5' end of the gene in control condition but was significantly lower in the Mago KD. These results thus suggest that the pre-EJC might be specifically required to maintain pausing rather than initiating it"

By

“During HS, Pol II occupancy rose dramatically and the extent of this increase was similar in control versus Mago KD, suggesting that Mago has no impact on transcription initiation. However, during recovery after HS, Pol II occupancy remained high at the 5’ end of the gene in control condition but was significantly lower in the Mago KD. These results thus further suggest that the pre-EJC is specifically involved in the control of Pol II pausing rather than in transcription initiation”

5. Lines 221-224: The authors again incorrectly describe their findings from Suppl. Fig. 6g in this sentence. There seems to be significant difference in Mago-HA ChIP signal upon Cwc22 KD or isoginkgetin treatment as compared to control. The differences are particularly prominent in the case of intron-containing genes. Perhaps, there could be some cross-talk between transcription and splicing dependent EJC factor recruitment mechanisms.

We agree with the reviewer that we could stress further the fact that splicing inhibition does affect Mago binding at promoters of intron-containing genes.

We modified this sentence:

“Further, the association of Mago-HA with the TSS was not substantially affected by the depletion of the spliceosome-associated factor CWC22 or by treatment with a splicing inhibitor, in particular for intronless genes. “

by

Further, the association of Mago-HA with the TSS was not substantially affected by the depletion of the spliceosome-associated factor CWC22 or by treatment with a splicing inhibitor on intronless genes. Nevertheless, applying the same condition on intron-containing genes significantly reduced Mago binding, suggesting that splicing can contribute to Mago enrichment at promoters“

6. Line 226: It should be Suppl. Fig. 6h and not 5h.

This has now been fixed.

7. Suppl. Fig 7a: The authors should include a sentence on why they see a reduction in Pol II occupancy on 3’ end of piwi, Crk3 and BBS8 genes but such a change is not seen in Luciferase construct (Fig. 3h).

The decrease of Pol II occupancy at 3’ end of piwi, Crk3 and BBS8 is due to increased pausing. This effect was not observed in Luciferase construct presumably because of sonication biases, given that the Luciferase construct is smaller than the other cloned genes. In addition, Pol II occupancy at 3’ end is relatively low, already in WT condition, which can generate variability by qPCR. As these data can be misleading we decided to

remove the analysis at 3' end of the genes and keep only the more reliable data from the 5' end.

8. Suppl. Fig 7b is not referenced in the main text.

This has now been included, see point 11

9. Suppl Fig. 8b is similar (the same?) as Suppl Fig. S6k. However, the former is labeled to show "pre-EJC bound" whereas the latter shows "Mago bound".

This has now been rectified and indicates "Mago-bound" in all the panels.

10. Line 346: Should the reference be to Suppl Fig. 9c instead of Suppl Fig. 7c?

Yes, this has now been fixed.

11. Line 349: Should the reference be to Suppl Fig. 7b instead of Suppl Fig. 6b?

This has been corrected.

Reviewer #2 (Remarks to the Author):

The authors have thoroughly revised the manuscript and answered all the questions raised. I recommend this manuscript to be accepted.

Reviewer #3 (Remarks to the Author):

Authors have answered most of the key questions raised by reviewers. However, I believe several points should be made clear for data presentation before publication.

1. It is not conventional to draw error bars from two biological data points. This applies to Figure 6e and most of meta gene plots comparing control KD and mago(h) KDs. Please display the standard deviation as confidence intervals in selected single libraries of each KD condition. Data deviation between two replicates should be shown in a different format if necessary.

All the metagene profiles indicate standard error of mean for all the expressed or Pol II bound genes, within the respective knock down condition after replicate averaging. Now the legend is modified accordingly to clearly indicate how the standard error of mean was derived in the metagene plots. The qPCR in Figure 6e is now reanalyzed to indicate 95% confidence interval from the mean.

2. Please provide all the original western blot images in a separate supplementary file.

We have now provided all the original western blot images in a separate supplementary file.

3. It is still not clear how CHIP-seq data are normalised. How can libraries be normalised both by spike-in and input? Please elaborate the method section rather than just mentioning which tools were used.

The spike-in normalization was done according to the Orlando et al 2014. All the libraries (including input) were calibrated using a normalization factor; defined as the number of reads mapped to the yeast genome (used as a reference/non-test genome) per million of reads mapped to the *Drosophila* genome. Once the libraries were calibrated according to the respective normalization factor, the enrichment was computed for every condition against their respective input.

Orlando, D.A. et al. Quantitative CHIP-Seq normalization reveals global modulation of the epigenome. *Cell Rep* **9**, 1163-70 (2014).

4. Fig. 2a-c and S6a, c: no genome-wide profile of RnpS1-HA CHIP is shown.

This has been included now in Fig 2a and b.

5. Please explain what x and y axes indicate and what individual dots represent in Figure S2c.

The figure S2c is correlation plot between the replicates of Pol II CHIP-Seq of indicated conditions. Each dot in the plot corresponds to one individual bin of the genome, and X and Y-axis corresponds to the strength of the signal in replicate 1 and replicate 2 respectively. Based on the correlation between the signal strength across all individual bins in the genome (indicated by the cluster of dots) between the replicates a Pearson correlation coefficient is computed, as indicated in the plot.

6. Box plots displaying the polymerase release ratio are difficult to read. Please use the full term whenever space is allowed and consider using "paused" and "less paused" with a triangle indicator instead of using low to high or quar. 1 to 4.

Now we have systematically modified the labeling of the plots to indicate "Polymerase Release Ratio" and also show the quartiles by a triangle indicator, as suggested.